# A deep-learning-based framework for identifying and localizing multiple abnormalities and assessing cardiomegaly in chest X-ray

Weijie Fan [1,8], Yi Yang [2,8], Jing Qi[2,8], Qichuan Zhang[1], Cuiwei Liao[1], Li Wen[1], Shuang Wang[1], Guangxian Wang[3], Yu Xia[4], Qihua Wu[5], Xiaotao Fan[6], Xingcai Chen[2], Mi He[2], JingJing Xiao[7], Liu Yang[1], Yun Liu[1], Jia Chen[1], Bing Wang[1], Lei Zhang[1], Liuqing Yang[1], Hui Gan[1], Shushu Zhang[1], Guofang Liu[1], Xiaodong Ge[1], Yuanqing Cai[1], Gang Zhao[1], Xi Zhang[1], Mingxun Xie[1], Huilin Xu[1], Yi Zhang[1], Jiao Chen[1], Jun Li[1], Shuang Han[1], Ke Mu[1], Shilin Xiao[1], Tingwei Xiong[1], Yongjian Nian [2] ✉ & Dong Zhang [1] ✉

Accurate identification and localization of multiple abnormalities are crucial steps in the interpretation of chest X-rays (CXRs); however, the lack of a large CXR dataset with bounding boxes severely constrains accurate localization research based on deep learning. We created a large CXR dataset named CXR-AL14, containing 165,988 CXRs and 253,844 bounding boxes. On the basis of this dataset, a deep-learning-based framework was developed to identify and localize 14 common abnormalities and calculate the cardiothoracic ratio (CTR) simultaneously. The mean average precision values obtained by the model for 14 abnormalities reached 0.572-0.631 with an intersection-over-union threshold of 0.5, and the intraclass correlation coefficient of the CTR algorithm exceeded 0.95 on the held-out, multicentre and prospective test datasets. This framework shows an excellent performance, good generalization ability and strong clinical applicability, which is superior to senior radiologists and suitable for routine clinical settings.

Chest X-ray (CXR) technology has become the initial imaging examination method for chest abnormalities because of its low cost and simple operating procedure[1]. The large number of CXRs generated worldwide are interpreted individually by radiologists, which requires considerable time and effort and increases the missed diagnosis and misdiagnosis rates[2]. The accurate automatic identification and localization of abnormalities in CXRs can effectively reduce the workload of radiologists and improve their diagnostic efficiency.

Previous studies have confirmed that deep learning can help radiologists efficiently interpret CXRs[3]. A deep learning-based

[1]Department of Radiology, Second Affiliated Hospital, Army Medical University, Chongqing 400037, P. R. China. [2]Department of Digital Medicine, School of Biomedical Engineering and Imaging Medicine, Army Medical University, Chongqing 400038, P. R. China. [3]Department of Radiology, People's Hospital of Banan, Chongqing Medical University, Chongqing 401320, P. R. China. [4]Department of Radiology, Xishui hospital of Traditional Chinese Medicine, Zunyi of Guizhou province 564600, P. R. China. [5]Department of Radiology, People's Hospital of Nanchuan, Chongqing 408400, P. R. China. [6]Department of Radiology, Fengdu People's Hospital, Chongqing 408200, P. R. China. [7]Department of Medical Engineering, Second Affiliated Hospital, Army Medical University, Chongqing 400037, P. R. China. [8]These authors contributed equally: Weijie Fan, Yi Yang, Jing Qi. ✉e-mail: yjnian@tmmu.edu.cn; hszhangd@tmmu.edu.cn

classification model was developed to screen abnormal CXRs in a previous study;[4] however, it would be more useful for radiologists if abnormalities in CXRs could be automatically identified. A disease-specific deep model was developed to detect certain chest diseases, such as nodules[5,6], tuberculosis[7], pneumothorax[8] or pneumonia[9]. However, it may not ultimately be beneficial to the clinician's interpretation because several coexisting abnormalities are commonly visible on CXR in actual clinical practice. Several studies[10,11] have developed deep classification models for multiple abnormalities using public CXR datasets, but these models only provide classification results for each CXR without any localization information. A few studies[12,13] have tried to localize abnormalities with the heatmaps generated by deep classification models. Nonetheless, heatmaps were only used to show which parts of a given CXR led the model to its final classification decision, so they could not strictly predict the standard bounding box of each abnormality[14]. In fact, to professionally realize the identification and localization of multiple abnormalities in CXRs, it is necessary to employ an object detection network from the field of computer vision based on a large CXR dataset with category and localization labels (bounding boxes) for each abnormality. Yongwon's study[15] developed an eDense You Only Look Once (YOLO) model for five CXR abnormalities with only 4,634 lesion masks, but its purpose was only to evaluate the reproducibility of the model. Nguyen[16] created a small CXR dataset called VinDr-CXR, and two studies[17,18] developed models based on this dataset to localize multiple lesions; however, it is difficult to achieve high localization performance due to the extremely limited number of bounding boxes. Notably, the lack of a large dataset with ground-truth (GT) bounding boxes significantly hampers the ability to accurately identify and localize multiple abnormalities in CXRs[14].

In addition, the size of the heart shadow should also be observed by radiologists in a CXR. The cardiothoracic ratio (CTR) is often used to assess the degree of cardiomegaly in clinical practice;[19] however, its manual calculation is relatively time-consuming.

In this work, we constructed a large CXR dataset named CXR-AL14 with bounding boxes for 14 abnormalities. Based on the CXR-AL14 dataset, a deep learning-based framework was developed for the identification and localization of 14 abnormalities and the simultaneous calculation of the CTR (Fig. 1). Finally, an intelligent diagnosis system was developed according to this framework to assist radiologists in more efficiently interpreting CXRs.

## Results

### Creation of the CXR-AL14 dataset

The creation of the CXR-AL14 dataset was a very large project. After the screening process of the original CXRs (Supplementary Figure 1), the CXR-AL14 dataset was finally constructed with the help of a human-in-the-loop approach. To ensure the accuracy of annotation, we formulated annotation principles for each abnormality, and the annotation of each CXR was corrected by at least one senior radiologist and one expert radiologist (see Methods). The general information and detailed distribution of all abnormalities in the CXR-AL14 dataset can be seen in Table 1. Except for the atelectasis (358 GT bounding boxes), the number of GT bounding boxes for the other 13 abnormalities were all larger than 2,998. The abnormality with the largest number of GT bounding boxes was nodules, up to 45,977. To the best of our knowledge, the CXR-AL14 dataset is the largest CXR dataset with bounding boxes for 14 common abnormalities. Based on the dataset, the YOLOX model was trained and tuned for the identification and localization of multiple abnormalities at a ratio of 9:1, and evaluated on

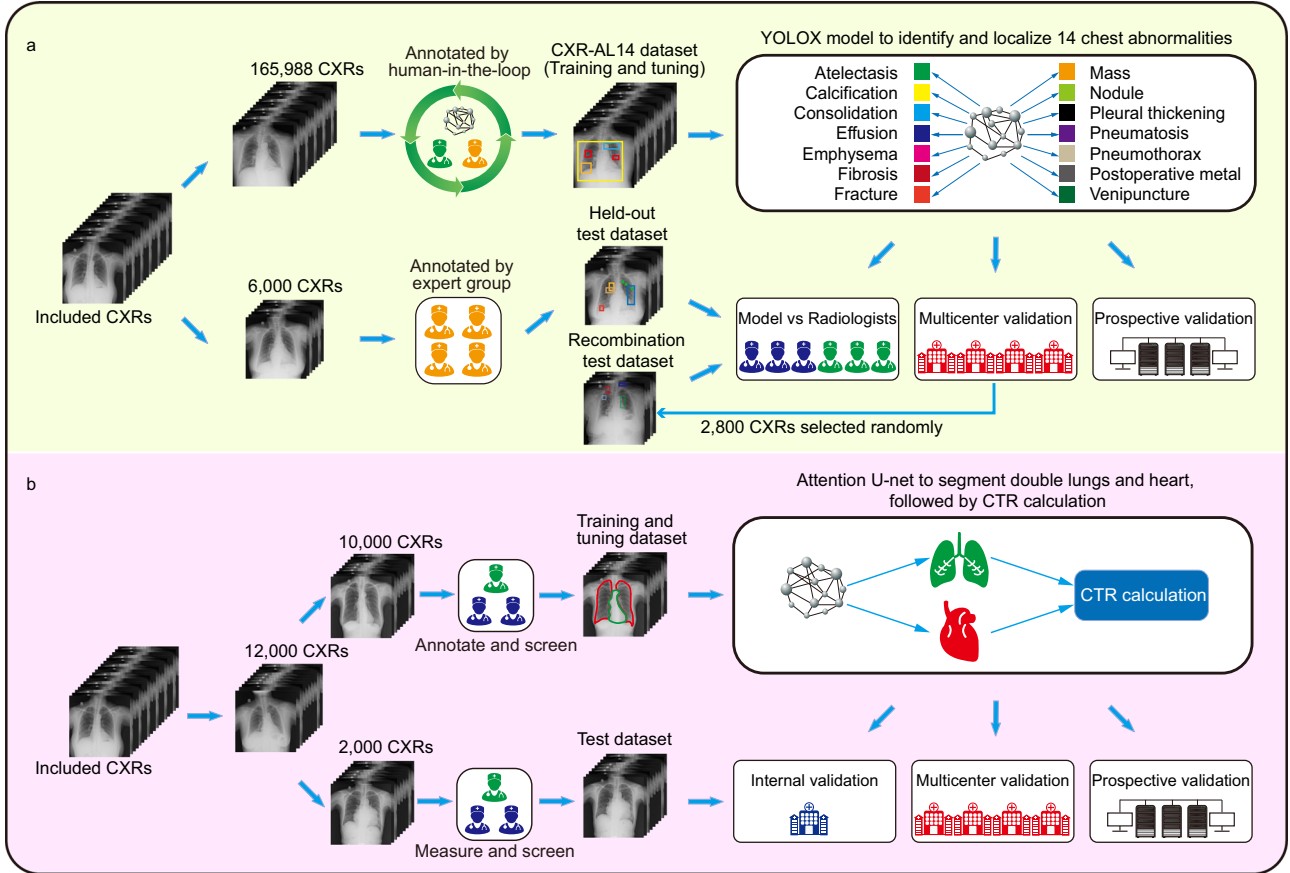

**Fig. 1 | The flowchart of the study. a** Development and validation of the YOLOX model for the identification and localization of 14 chest abnormalities. **b** Development and validation of an algorithm for calculating the CTR to quantitatively assess cardiomegaly. CTR: cardiothoracic ratio.

**Table 1 | The general information and distribution of all abnormalities in the CXR-AL14 dataset**

|  | Training dataset | Tuning dataset | CXR-AL14 dataset |
|---|---|---|---|
| CXR images | 149,425 | 16,563 | 165,988 |
| Abnormal CXRs | 92,620(61.984%) | 10,284(62.090%) | 102,904(61.995%) |
| No finding CXRs | 56,805(38.016%) | 6,279(37.910%) | 63,084(38.005%) |
| Patients | 130,478 | 14,490 | 144,968 |
| Age (mean ± sd) | 53.44 ± 15.20 | 53.43 ± 15.25 | 53.44 ± 15.20 |
| Gender (males, %) | 83,584(55.937%) | 9370(56.572%) | 92,954(56.000%) |
| Atelectasis | 317(0.139%) | 41(0.162%) | 358(0.141%) |
| Calcification | 30,020(13.140%) | 3334(13.134%) | 33,354(13.140%) |
| Consolidation | 19,103(8.362%) | 2163(8.521%) | 21,266(8.378%) |
| Effusion | 39,781(17.413%) | 4330(17.057%) | 44,111(17.377%) |
| Emphysema | 21,240(9.297%) | 2350(9.257%) | 23,590(9.293%) |
| Fibrosis | 15,043(6.585%) | 1644(6.476%) | 16,687(6.574%) |
| Fracture | 11,422(5.000%) | 1210(4.767%) | 12,632(4.976%) |
| Mass | 2701(1.182%) | 297(1.170%) | 2998(1.181%) |
| Nodule | 41,438(18.138%) | 4539(17.881%) | 45,977(18.112%) |
| Pleural thickening | 6570(2.876%) | 823(3.242%) | 7393(2.912%) |
| Pneumatosis | 6229(2.727%) | 724(2.852%) | 6953(2.739%) |
| Pneumothorax | 6350(2.779%) | 714(2.813%) | 7064(2.783%) |
| Postoperative Metal | 17,845(7.811%) | 2084(8.210%) | 19,929(7.851%) |
| Venipuncture | 10,400(4.552%) | 1132(4.459%) | 11,532(4.543%) |
| Total | 228,459 | 25,385 | 253,844 |

(Source data are provided as a Source Data file.).

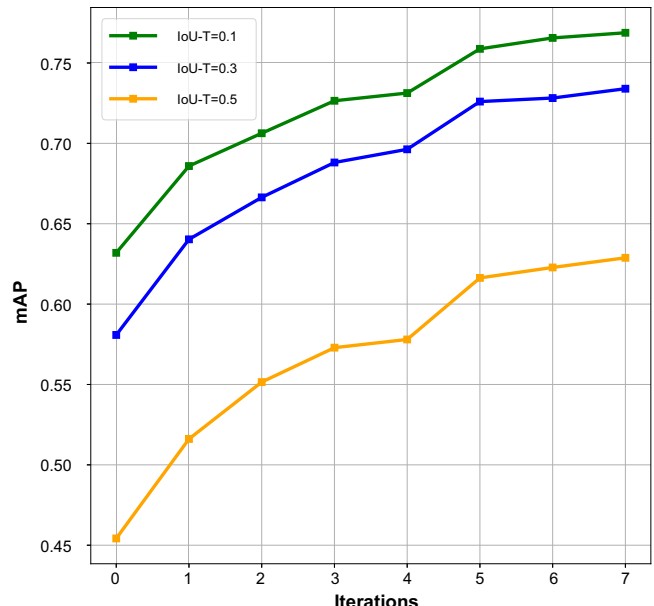

**Fig. 2 | Performance of the updated model after each iteration with different IoU-Ts.** mAP: mean average precision, IoU-T: intersection over union threshold. (Source data are provided as a Source Data file.).

held-out, multicentre and prospective test datasets. Manufacturer and device information for each dataset is displayed in Supplementary Table 1.

The interreader and intrareader variability based on six experts' annotations were assessed in this study. It is well known that the larger the intersection over union (IoU) value is, the higher the overlap rate between two bounding boxes and the better their consistency[20]. The assessment shows that the average intrareader IoU value of each abnormality ranged from 0.773 to 0.992, and the average interreader IoU value of each abnormality ranged from 0.698 to 0.968. More detailed information can be found in Supplementary Table 2. In addition, we found that IoU=0 for a few abnormalities. This occurred in two situations. One was that certain abnormalities were annotated by one expert but not by another expert, and the other was that certain abnormalities were annotated differently by two experts. We further counted the number of bounding boxes with IoU=0 for each abnormality and found that the maximal number was 14 in the inter-reader study (<0.023 per CXR) and the maximal number was 6 in the intrareader study (<0.01 per CXR). Detailed information can be found in Supplementary Table 3. These results demonstrated the good interreader and intrareader agreement of annotations.

The performance achieved by the updated deep model after each iteration was separately evaluated on the held-out test dataset, as shown in Fig. 2. The results demonstrated that as the number of iterations increased, the number of annotated CXRs also increased, and the performance of the updated model gradually improved. Note that the model with 0 iterations corresponds to the preliminary model. After seven iterations, the construction of the CXR-AL14 dataset was completed, and the updated model after the seventh iteration was the YOLOX model in the proposed framework, which achieved the highest performance. The results also demonstrated that the human-in-the-loop approach was effective for the annotation of the training dataset.

## Result of the 5-fold cross-validation

Five-fold cross-validation was performed on the CXR-AL14 dataset, and five models were generated. The performance results of the five models in the 5-fold cross-validation with an IoU-T of 0.5 are given in Supplementary Fig. 2, and the mean average precision (mAP) values of the five models were 0.591, 0.620, 0.626, 0.616 and 0.610, respectively. Moreover, the five models were further tested on the held-out test dataset, the corresponding results with an IoU-T of 0.5 are illustrated in Supplementary Fig. 3, and the corresponding mAPs were 0.599, 0.612, 0.619, 0.608 and 0.605, respectively. From these results, it was clear that all five models trained on the CXR-AL14 dataset had similar performance, suggesting that the performance of the model trained on the CXR-AL14 dataset would possess good stability and repeatability.

## Performance of the YOLOX model on the held-out test dataset

The preliminary validation of the performance of the YOLOX model was performed on the held-out test dataset, which was independent of the CXR-AL14 dataset (general information shown in Supplementary Table 4). When the IoU threshold (IoU-T) was set to 0.5, 0.3 and 0.1, the mAP values of 14 abnormalities achieved by the YOLOX model reached 0.629, 0.734 and 0.769, respectively. Except for that of atelectasis, the AP values of the other 13 abnormalities were all greater than 0.348 with three IoU-Ts. Note that the AP value of venipuncture was the highest and was higher than 0.9 with three IoU-Ts (Fig. 3). The PR curve of each abnormality obtained by the YOLOX model with different IoU-Ts can be found in Supplementary Fig. 4. Three examples interpreted by the YOLOX model for the identification and localization of multiple abnormalities and their corresponding GT bounding boxes are shown in Fig. 4.

In addition, when the IoU-T was set to 0.5, 0.3 and 0.1, the mAPs of the Faster R-CNN model were 0.271, 0.423, and 0.508, respectively, while the mAPs of the RetinaNet model were 0.506, 0.632, and 0.678, respectively. More details about the AP of each abnormality determined by the two models are shown in Supplementary Table 5. The performance of the YOLOX model was also compared with that of the Faster R-CNN model and the RetinaNet model, as shown in Supplementary Fig. 5. We can easily find that the performance of the YOLOX model was superior to that of the other two models.

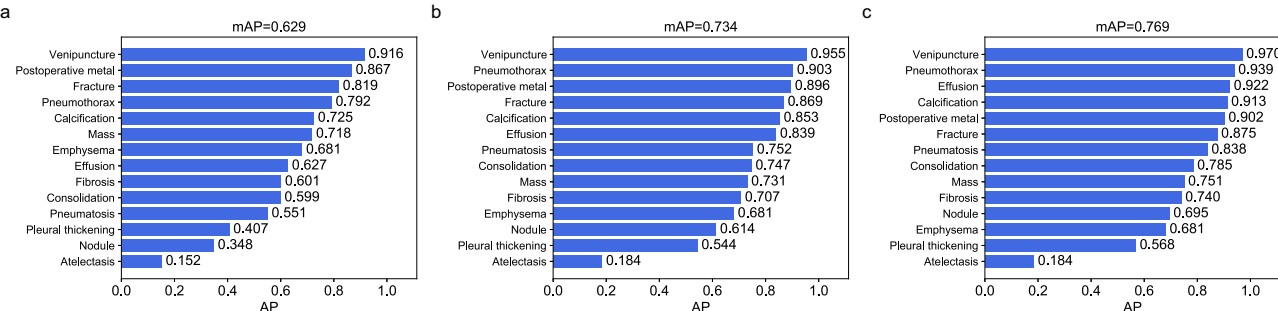

**Fig. 3 | Performance of the YOLOX model on the held-out test dataset with different IoU-Ts. a** AP values of 14 abnormalities with an IoU-T of 0.5, **b** AP values of 14 abnormalities with an IoU-T of 0.3, **c** AP values of 14 abnormalities with an IoU-T of 0.1. AP: average precision, IoU-T: intersection over union threshold. (Source data are provided as a Source Data file.).

## Multicentre validation

To test the generalizability and robustness of the YOLOX model, another four completely independent external validation datasets were collected from primary hospitals in districts or counties (general information shown in Supplementary Table 4). When the IoU-T was set to 0.5, the YOLOX model achieved excellent performance in terms of the identification and localization of 14 abnormalities on four multicentre test datasets, and the mAP values were 0.586, 0.580, 0.605 and 0.572 (Supplementary Fig. 6). When the IoU-T was set to 0.3 or 0.1, the mAP of each external dataset also increased (Supplementary Figs. 7 and 8). The highest mAP of 0.747 was obtained with an IoU-T of 0.1 on external dataset A whose CXRs form People's Hospital of Banan, and this value was slightly lower than that obtained by the YOLOX model (0.769). The average precision for each external dataset was 0.659, 0.651, 0.651 and 0.588, while the average recall values were 0.624, 0.621, 0.651 and 0.643, respectively. Encouragingly, the average F1-scores achieved by the model were all over 0.60. The precision, recall and F1-scores obtained for each abnormality with different IoU-Ts on the multicentre test datasets can be found in Supplementary Tables 6-8. For different abnormalities, the AP values on different multicentre validation datasets had a certain fluctuation, which is caused by the different data distributions, but the overall generalization ability of the YOLOX model is eximious.

Furthermore, to demonstrate the accuracy of the YOLOX model in terms of localizing abnormalities, we calculated the average IoU value for each abnormality (in comparison with the GT bounding boxes annotated by the expert radiologists) on the external test dataset (a total of 10,945 CXRs from four multicentres). The average IoU values ranged from 0.458 to 0.917; more details can be found in Supplementary Table 9. It is easy to see that the YOLOX model achieved excellent performance with respect to the identification and localization of multiple abnormalities.

## Performance comparison between the YOLOX model and radiologists

The performance comparison between the YOLOX model and radiologists was carried out on the held-out test dataset and the recombination test dataset. Three junior radiologists and three senior radiologists were asked to independently annotate all CXRs in the above two test datasets. On the held-out test dataset, when the IoU-T was set from 0.1 to 0.5 with a step of 0.1, it was clear that the F1-scores of the 14 chest abnormalities achieved by the YOLOX model were more stable than those of the six radiologists (Fig. 5). We can see the PR curves of the 14 abnormalities and the results obtained by the six radiologists with an IoU-T of 0.5 in Fig. 6. The mean precision, recall and F1-score obtained by the YOLOX model were 0.693, 0.651 and 0.665, respectively, which were higher than those of the best senior radiologist, whose values were 0.626, 0.547 and 0.577, respectively (Supplementary Table 10). More details regarding the performance of

each radiologist are shown in Supplementary Table 11. The results obtained with the other IoU-Ts are shown in Supplementary Figs. 9-10 and Tables 12-15. Overall, the YOLOX model was superior to the six radiologists in identifying and locating most abnormalities. However, the YOLOX model performed worse than all six radiologists in the identification of atelectasis and emphysema. From Supplementary Figs. 11 and 12, we can see that the YOLOX model outperforms the six radiologists in reducing missed diagnoses. We performed one-way analysis of variance (ANOVA) for the F1-scores achieved by the YOLOX model, junior radiologists, and senior radiologists, and used the least significant difference (LSD) method for pairwise comparisons between groups (with an IoU-T of 0.5). The results (Supplementary Tables 16-17) showed that the performance of the YOLOX model was significantly better than that of the junior radiologists ($P = 0.034$).

Similarly, on the recombination test dataset, the YOLOX model also showed good performance, which was superior to that of the senior radiologists. With an IoU-T of 0.5, the mean precision, recall and F1-score obtained by the YOLOX model were 0.640, 0.639 and 0.632, respectively, which were higher than those of the best senior radiologist, whose values were 0.632, 0.555 and 0.586, respectively (Supplementary Tables 18-19). From Supplementary Fig. 13, we found that the performance of the YOLOX model was better than that of the radiologists for most abnormalities. However, the performance of the YOLOX model for atelectasis and emphysema was also lower than that of six radiologists. The performance achieved on the recombination test dataset was similar to the results obtained on the held-out test dataset. ANOVA and the LSD method for pairwise comparisons between groups were further performed and showed that the differences between the groups were not statistically significant (Supplementary Tables 20-21). However, the mean F1-score achieved by the YOLOX model was higher than those of the six radiologists.

## Prospective validation

To further validate the clinical applicability of the YOLOX model, a prospective test dataset was constructed. The mAP values obtained by the YOLOX model reached 0.631, 0.731 and 0.753 with IoU-T of 0.5, 0.3 and 0.1, respectively. (Supplementary Fig. 14). It is not difficult to find that the model still obtains high performance, and its mAP value with IoU-T of 0.5 even was the highest in all test datasets. However, regardless of how the IoU-T changed, the AP value of atelectasis was always lowest (0.222). The precision, recall and F1-score values obtained for each abnormality with different IoU-Ts on the prospective test dataset can be found in Supplementary Tables 6-8.

## False-positive (FP) findings by the YOLOX model on each test dataset

The number of FP findings in each CXR is an important index that reflects how much additional attention and effort would be needed in the CXR interpretation workflows. We calculated the number of FP

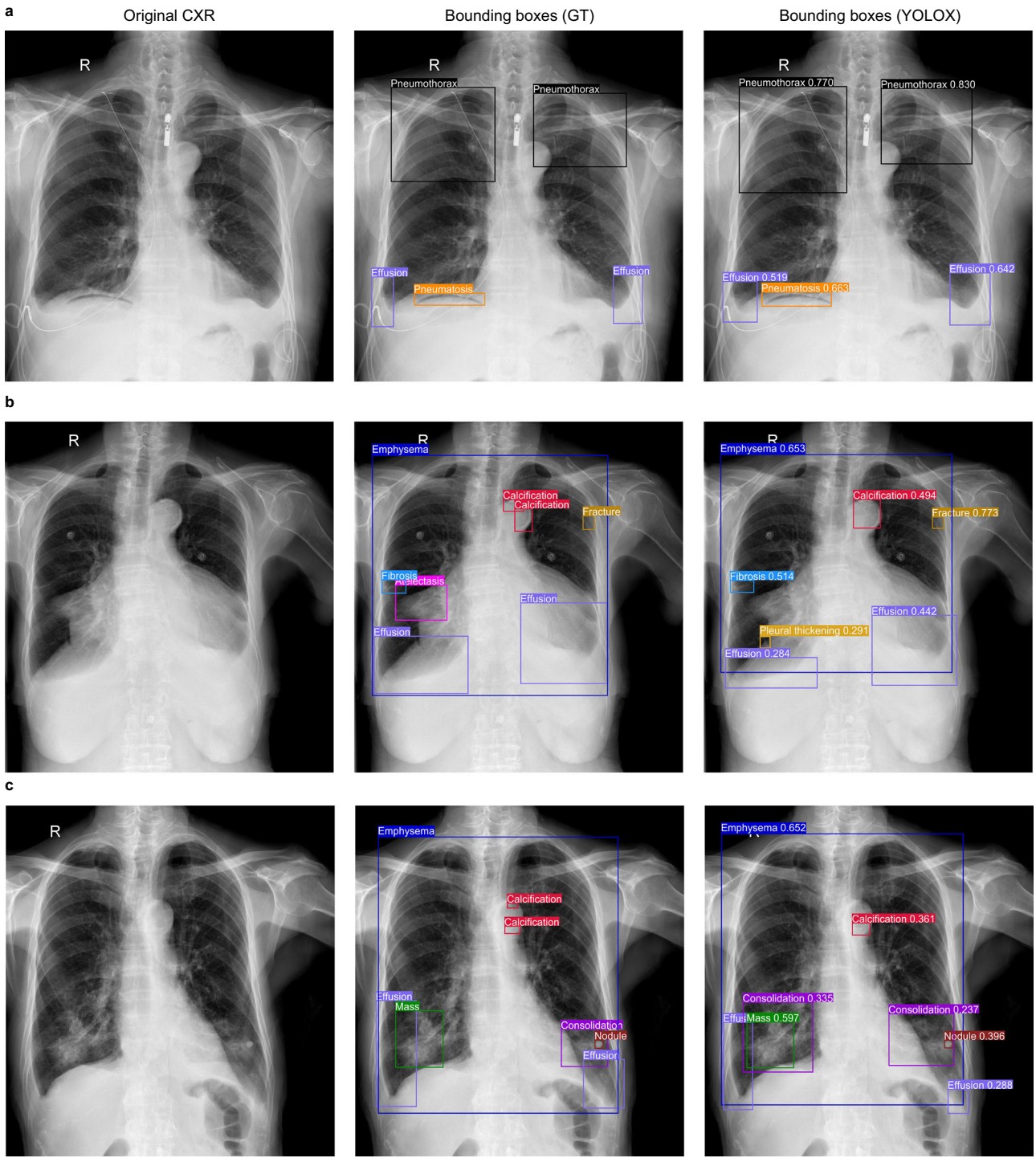

**Fig. 4 | Three examples of the YOLOX model for the identification and localization of multiple abnormalities in CXRs. a** In the case A, the category and localization of bounding boxes predicted by the YOLOX model were highly coincided with the GT bounding boxes; **b** In the case B, the YOLOX model missed atelectasis in the right lung hilar region and merged two separate calcifications on the aortic arch into a single calcification; **c** In the case C, The YOLOX model missed microcalcifications on the aortic arch and additionally diagnosed consolidation around the right lung mass. However, the other bounding boxes predicted by YOLOX model in case B and C were also highly coincided with the GT bounding boxes. GT: ground-truth.

findings for each abnormality in each CXR contained in all test datasets with different IoU-T values, and the results are shown in Supplementary Table 22. Note that nodules accounted for the largest proportion of FP findings among all abnormalities (e.g., 53.23% in external test dataset B with an IoU-T of 0.3). However, the average number of FP findings of all abnormalities in each CXR did not exceed one with an IoU-T of 0.5. Thus, we believe that the YOLOX model has good performance in controlling FP findings and will not increase the workload of radiologists.

### Stress test of the YOLOX model
Figure 7 shows the performance change curves of the YOLOX model under different brightness or contrast. Notably, small changes in brightness or contrast had little effect on the performance of the

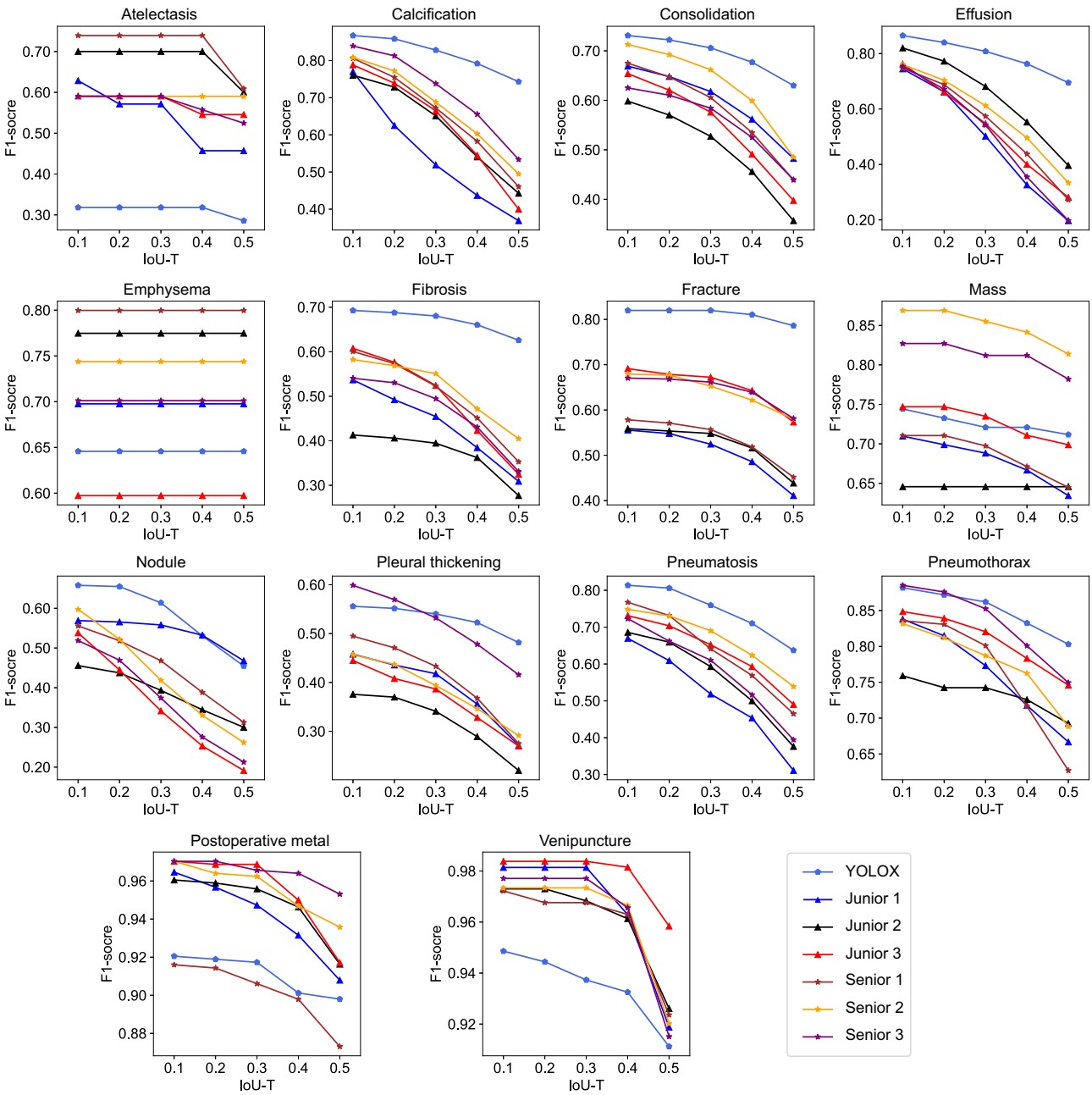

**Fig. 5 | The F1-scores for 14 chest abnormalities produced by the YOLOX model and six radiologists with different IoU-Ts.** IoU-T: intersection over union threshold. (Source data are provided as a Source Data file.).

YOLOX model; in contrast, large changes in brightness or contrast had marked impacts on it. However, in routine clinical practice, there are few CXRs with particularly poor brightness and contrast. In addition, the brightness and contrast levels of the CXRs in each test dataset may be different because the CXRs were generated by various X-ray equipment. However, the YOLOX model achieved good performance on each test dataset. All the above suggest that the model had good generalizability.

### Validation of the CTR calculation process

A certain number of CXRs were randomly selected from different test datasets to validate the CTR calculation algorithm (Supplementary Tables 23 and 24). The reference standards of each CTR were measured collaboratively by two junior radiologists and one senior radiologist (see Methods). The mean values of the differences between the values calculated by the proposed CTR calculation algorithm and the

reference standards were 0.011 on the held-out test dataset and 0.005, 0.002, 0.012 and 0.009 on each external dataset, respectively (Fig. 8a-e). In the prospective validation, the mean value of the difference was 0.002 (Fig. 8f). Note that the intraclass correlation coefficient (ICC) values between the calculated values and the reference standards were all over 0.95 on all test datasets (Fig. 8).

## Discussion

In our study, a tremendous CXR-AL14 dataset containing 165,988 CXRs (102,904 abnormal CXRs and 63,084 "No finding" CXRs) with 253,844 GT bounding boxes for 14 chest abnormalities was created. To the best of our knowledge, this CXR dataset possesses the largest number of GT bounding boxes in the world. Then, we developed a deep learning-based framework that could identify and localize 14 chest abnormalities and simultaneously calculate the CTR. The diagnostic efficiency of the proposed framework was superior to that of senior radiologists.

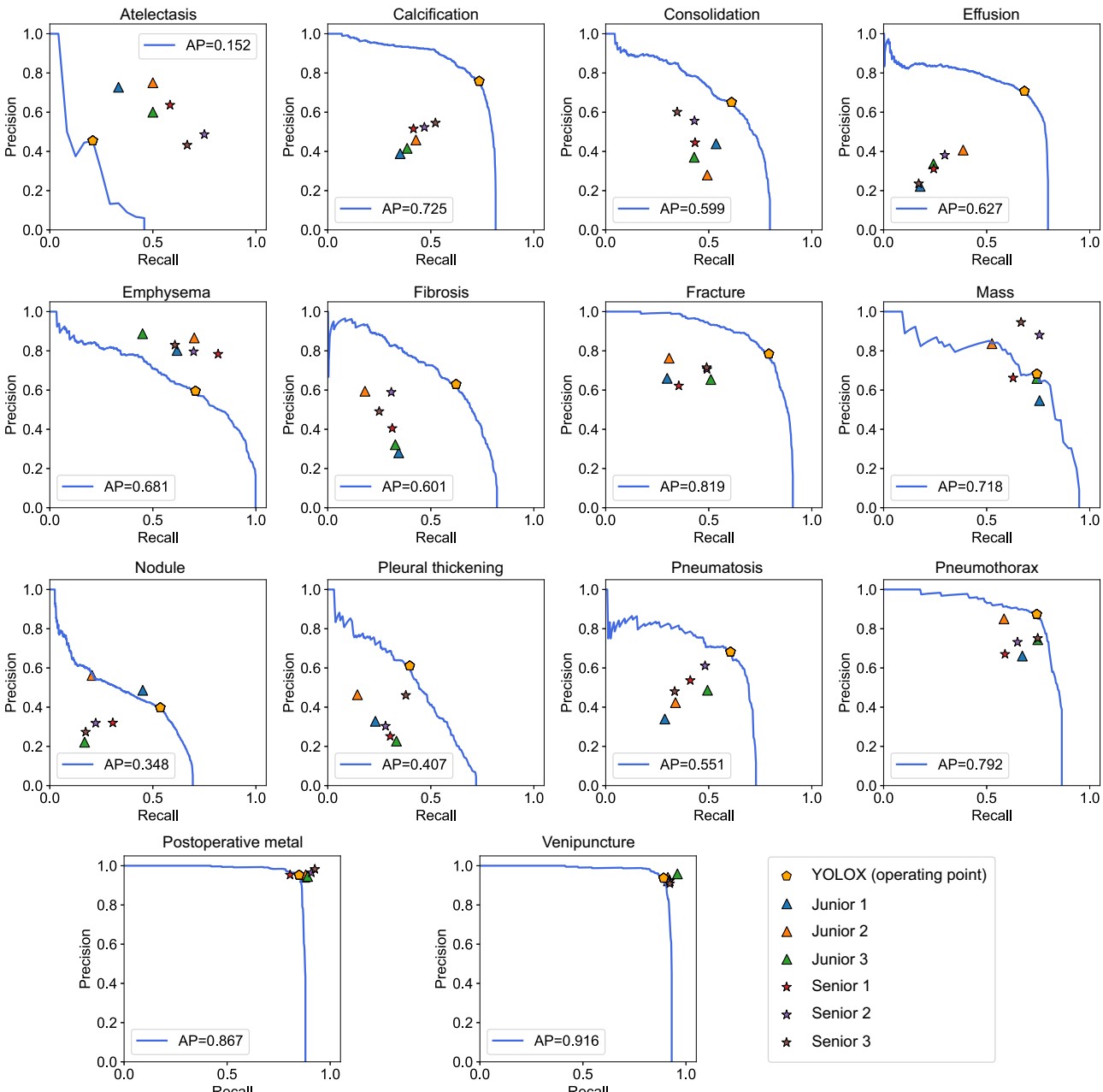

**Fig. 6 | Performance comparison between the YOLOX model and radiologists on the held-out test dataset with an IoU-T of 0.5.** The triangles represent the junior radiologists, the pentacles represent the senior radiologists, and the pentagon represents the model operation point. AP: average precision. IoU-T: intersection over union threshold. (Source data are provided as a Source Data file.).

Multicentre and prospective validations demonstrated that the framework has good generalization performance and clinical applicability.

Accurate identification and localization based on deep learning rely heavily on a large number of CXRs with GT bounding boxes for multiple abnormalities. At present, some popular public CXR datasets, such as ChestX-ray14[21], MINIC-CXR[22], PadChest[23] and CheXpert[24], can be utilized in deep learning research. Nonetheless, the above CXR databases are imperfect and contain only category labels extracted from CXR reports via natural language processing (NLP). It is well known that there are errors in the labels obtained by NLP, and the inaccuracy rate may reach 14%[25]. Moreover, these public datasets lack localization information for each abnormality, which poses a barrier with respect to the exploration of accurate localization approaches.

Note that the well-known ChestX-ray14 dataset contains only 984 bounding boxes, and the limited training data are not sufficient for developing a deep localization model. Nguyen[16] created a CXR dataset called the VinDr-CXR dataset, which contains 18,000 CXRs (5,343 abnormal images) with 17,367 bounding boxes. However, the numbers of bounding boxes for most abnormalities are less than 1000, and some of them are even less than 100. Such a small number of bounding boxes makes it difficult to develop high-performance deep localization models, and the mAP was only 0.365 with an IoU-T of 0.4. It is notoriously expensive and difficult to annotate a large-scale CXR dataset with GT bounding boxes[26]. In our study, with the help of the human-in-the-loop approach, which can decrease the annotation burden for radiologists, the CXR-AL14 dataset containing a tremendous number of GT bounding boxes for 14 abnormalities was created, and the

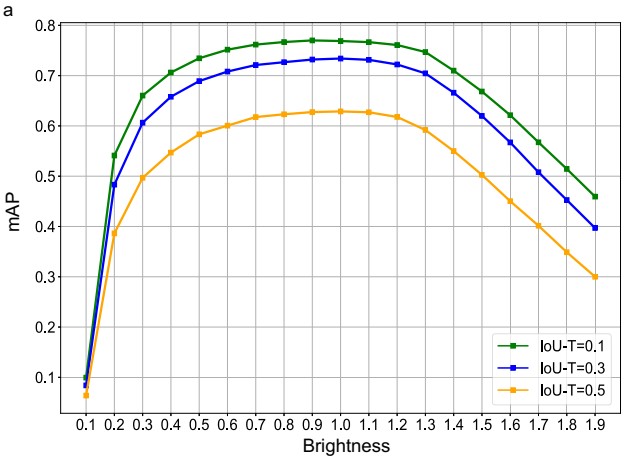
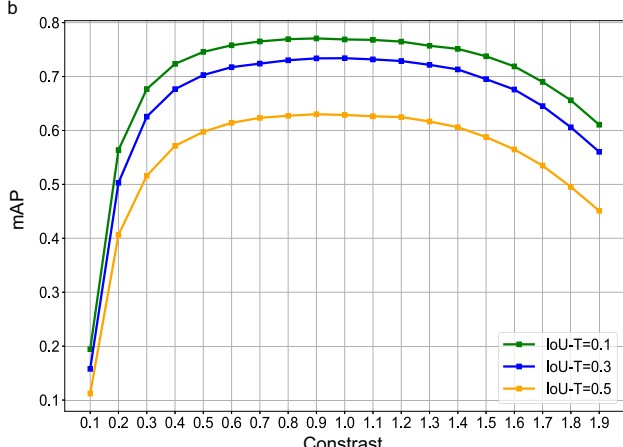

**Fig. 7 | Performance curves of the YOLOX model under different changes of brightness or contrast.** Set the brightness (**a**) or contrast (**b**) of all the CXRs in the held-out test dataset to 1.0, then negative and positive brightness or contrast changes with increments and decrements of 0.1 were tested. It was obvious that small changes in brightness and contrast had little effect on the performance of the YOLOX model. mAP: mean average precision, IoU-T: intersection over union threshold. (Source data are provided as a Source Data file.).

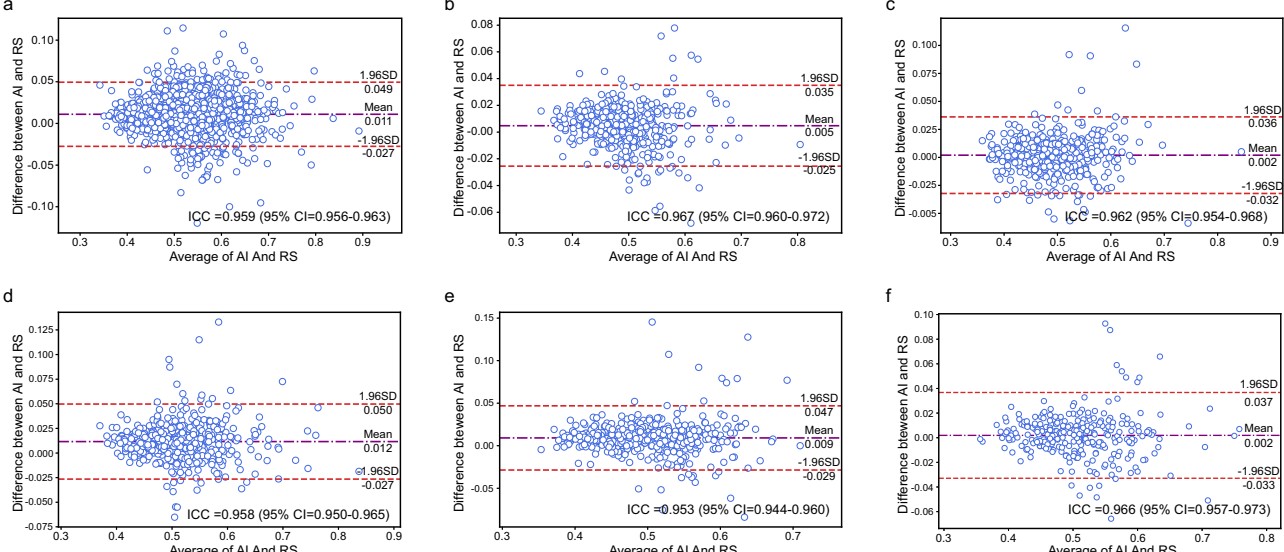

**Fig. 8 | Bland–Altman analysis plot between the results produced by the CTR algorithm and RS. a** Internal test dataset, **b** External dataset A, **c** External dataset B, **d** External dataset C, **e** External dataset D, **f** prospective validation. RS: Reference Standard. ICC: Interclass correlation coefficient. (Source data are provided as a Source Data file.).

numbers of bounding boxes for most abnormalities exceeded 10,000, enabling deep learning research on CXRs to move forward. The CXR-AL14 dataset can provide considerable data support for deep learning-based localization tasks, making the concept of simultaneous localization of multiple abnormalities on CXRs a reality. Most previous studies only used classification networks based on public CXR datasets to find abnormalities without accurate localization information. For the AI imaging diagnosis tool, the interpretation of our framework is completely consistent with the workflow of radiologists and more suitable for routine clinical practice than that of previous models trained by classification networks. Moreover, our framework had good performance, and the mAP achieved by the YOLOX model reached 0.629 with an IoU-T of 0.5. The performance of the YOLOX model is obviously superior to that of the models trained on a small CXR dataset, such as the VinDr-CXR dataset[16].

For computer-aided diagnosis, the ultimate goal is to be able to identify and localize all the diseases present on CXR. Unfortunately, it is difficult to accurately make disease diagnoses using only CXRs in clinical work. Certain chest diseases may exhibit various radiological features, and different chest diseases may share similar radiological features. To make a good CXR report, radiologists must be aware of the clinical symptoms and laboratory examination results, in addition to comprehensively reviewing all radiological abnormalities in CXRs. Therefore, our framework focused on the identification and localization of various abnormal radiological signs (not the detection of diseases), which is of paramount importance in the interpretation of CXRs. In addition, before radiologists annotated CXRs, we developed annotation principles for each abnormality to reduce the difference in annotation by different radiologists. Therefore, we believe our framework has practical value in an actual clinical setting.

Although we formulated the annotation principles for each abnormality, it was still difficult to ensure that the sizes and localizations of the bounding boxes annotated by different radiologists were completely consistent due to the cognitive differences, small sizes or ill-defined edges of some abnormalities. With the increase in the IoU-T, the degree of overlap between the predicted bounding box and the GT box must be more stringent to achieve good performance. In our study, the performance of both the YOLOX model and the radiologists decreased as the IoU-T increased. In particular, the performance achieved for small abnormalities, such as lung nodules and mild effusion, decreased faster. While the bounding boxes of emphysema were large and well-defined, which led to a high degree of overlap with the GT bounding boxes, the performance achieved for emphysema was not affected by the IoU-T. It is encouraging that the F1-scores of most abnormalities obtained by the YOLOX model became more stable than those of the radiologists as the IoU-T changed.

In a performance comparison with radiologists, the overall performance of the YOLOX model was superior to that of all six radiologists even with an IoU-T of 0.5, especially for calcification, consolidation, effusion, fibrosis, fractures, pleural thickening, pneumatosis, and pneumothorax. The precision levels obtained for most abnormalities by the YOLOX model were higher than those of all six radiologists. Moreover, the recall values of most abnormalities achieved by the YOLOX model were higher than those of the radiologists. These results demonstrated that the developed YOLOX model can greatly reduce the missed diagnosis and misdiagnosis rates produced for lesions and has the potential to be applied in routine clinical practice. For the identification of emphysema, radiologists outperformed the YOLOX model because the imaging diagnosis process has multiple essential objectives, such as a widened thorax and intercostal space, sparse lung texture, and a flattened diaphragmatic crest. Note that the AP and F1-score values of atelectasis were lower than those of the six radiologists under each IoU-T because the number of GT bounding boxes contained in the CXR-AL14 dataset for atelectasis was limited, and these boxes were insufficient for achieving good performance. In addition, the tip of a venipuncture has a high opacity and is easy to identify, but it sometimes overlaps with the thoracic vertebra, interfering with the identification and localization of the model. In the case of postoperative metals, the YOLOX model usually misidentified body surface metals as postoperative metals. Therefore, the F1-scores of venipuncture and postoperative metal obtained by the YOLOX model were slightly lower than those obtained by radiologists.

The multicentre validation demonstrated that the mAP values obtained for all four centres exceeded 0.570 with an IoU-T of 0.5. The YOLOX model still maintains good performance on external datasets, although there were some differences in the distribution of the 14 abnormalities between the CXR-AL14 dataset and the external test datasets. We found that the proportion of nodules in external datasets A-D was higher than that in the CXR-AL14 dataset. In contrast, the proportion of pneumatosis, pneumothorax, postoperative metal and venipuncture was lower than that in the CXR-AL14 dataset, which may be due to the different hospital levels of this dataset. Thus, the distributions of the CXR abnormalities at four centres were different from that at our hospital. However, our results demonstrated that the distribution of abnormalities had little effect on the performance of the YOLOX model. The AP values and F1-scores of atelectasis on each test dataset were limited because of the small number of bounding boxes of atelectasis on the CXR-AL14 dataset. Moreover, the AP values and F1-scores of pneumatosis and effusion on the multicentre test datasets were lower than those on the held-out test dataset. In general, small abnormalities, blurred boundaries, and multiple scattered abnormalities are difficult to accurately identify and localize. In the multicentre test datasets, a small amount of pleural effusion was more common, and pneumatosis was usually small and scattered in the CXRs from multicentre datasets. Thus, the performance of the YOLOX model for localizing these two abnormalities was limited. The performance achieved for atelectasis was also not satisfactory on the prospective test dataset (only 6 bounding boxes for atelectasis in the dataset).

To our knowledge, there are currently two methods for calculating CTR. In one method, CTR is the ratio of the maximal horizontal cardiac diameter to the maximal horizontal thoracic diameter (inner edge of ribs/edge of pleura), and CTR > 0.50 is usually considered cardiomegaly. Several previous studies[27,28] have been conducted on the automatic calculation of the CTR by referring to the above method; However, in professional reference book[19], CTR is defined as the ratio of the cardiac diameter (the horizontal distance between the most rightward and most leftward margins of the cardiac shadow) to the thoracic diameter (the distance from the inner margin of the ribs at the level of the dome of the right hemidiaphragm), and CTR > 0.55 is usually considered to be cardiomegaly. The automatic calculation of CTR according to the former method only requires the segmentation of both lungs and the heart. However, for the latter method, the dome point of the right hemidiaphragm needs to be automatically identified in addition to the segmentation of both lungs and the heart. In our studies, the automatic calculation of CTR in the framework was developed by referring to the latter method. In addition, these two methods cannot be applied to anterior-posterior (AP) CXRs and bedside CXRs due to the amplification effect of the heart shadow in these two cases. It is worth noting that when the anatomical boundaries of the heart are invisible due to the presence of lesions near the heart shadow, the CTR cannot be calculated.

There were some limitations in our study. First, the CXRs in the created CXR-AL14 dataset came from a single hospital, and adding CXRs from other hospitals may improve the performance of the deep model. Second, the distribution of the numbers of different types of abnormalities was unbalanced, which reduced the performance achieved by the YOLOX model. Third, lateral CXRs, which may provide additional information for identifying abnormalities, were not included in the study.

In conclusion, we created a large CXR-AL14 dataset containing category and localization information approximately for 14 abnormalities. It is expected that the CXR-AL14 dataset will promote further localization research involving CXRs. Based on the created dataset, a framework was developed to identify and localize 14 abnormalities and simultaneously calculate the CTR. Internal and external validations demonstrated that our framework has the potential to be used in auxiliary CXR diagnoses, efficiently reducing the workloads of radiologists and improving their diagnostic efficiency.

## Methods

### Ethics and information governance

This study was approved by the Medical Ethics Committee of the Second Affiliated Hospital of Army Medical University (no. 2021-159-01, no. 2022-193-01 and no. 2023-123-01). All methods were implemented in accordance with the approved regulations and the Declaration of Helsinki. The CXR-AL14 dataset and held-out test dataset were collected retrospectively with a waiver granted for the requirement of informed consent (no. 2021-159-01). The multicentre validation in this study was approved by the Medical Ethics Committee of the principal investigator's hospital. The multicentre hospital retrospectively collected CXR data according to the approved experimental procedures (no. 2022-193-01), and informed consent was waived for this retrospective analysis. Moreover, the CXRs of all the retrospective datasets, including the CXR-AL14 dataset, held-out, and multicentre test datasets were de-identified to remove any patient-related information before collection. None of the authors participated in the data de-identification process. The prospective test dataset was collected prospectively in accordance with procedures approved by the hospital Ethics Committee (no. 2023-123-01) and

written informed consent was obtained from each participant. The CXRs of prospective test datasets were also de-identified before transfer to study investigators.

## Study design

The flowchart of the study is shown in Fig. 1. In brief, our study formulated two tasks. One was to create a large dataset (CXR-AL14) with high-quality annotations for the training and tuning of the YOLOX model to enable it to identify and localize 14 chest abnormalities. A held-out test dataset that was independent of the CXR-AL14 dataset and a recombination test dataset, were used to evaluate the performance of the YOLOX model and compare it with that of radiologists. Moreover, multicentre and prospective validations were conducted to evaluate the generalization of the YOLOX model (Fig. 1a). The other task was to propose a CTR calculation algorithm based on deep learning for the quantitative assessment of cardiomegaly. Similarly, internal, multicentre and prospective validations were performed to evaluate the performance of the proposed algorithm (Fig. 1b). Finally, the above two tasks were integrated into a framework that could assist radiologists in reviewing CXRs in clinical practice.

## Data collection

We consecutively collected 315,072 original CXRs from 159,996 patients at the Department of Radiology, Second Affiliated Hospital, Army Medical University (AMU), between August 2011 and December 2021. Lateral or anteroposterior CXRs, CXRs from patients under the age of eighteen, and duplicate CXRs were excluded from the study. After filtering, 171,988 CXRs from 150,914 patients remained. Then, 6,000 CXRs were randomly selected as a held-out test dataset. Finally, all 165,988 remaining CXRs were annotated by the human-in-the-loop approach to create the CXR-AL14 dataset. The CXR screening process can be found in Supplementary Fig. 1. All CXRs were taken by the following three devices: AXIOM Aristotle VX Plus (Siemens, Germany), DRX Evolution (Carestream Health, Canada) and DirectView DR7500 (Kodak, USA).

Multicentre validation datasets were randomly collected from four hospitals, namely, External dataset A (People's Hospital of Banan in Chongqing), External dataset B (Fengdu People's Hospital of Chongqing), External dataset C (People's Hospital of Nanchuan in Chongqing) and External dataset D (Xishui Hospital of Traditional Chinese Medicine in Guizhou). After filtering the data with the above exclusion criteria, 2,978, 2,633, 2,651 and 2,683 CXRs from each hospital were included in this study. Furthermore, 700 CXRs from each multicentre were randomly selected to construct the recombination test dataset (a total of 2800 CXRs) for a further performance comparison between the YOLOX model and radiologists.

Prospective test dataset was collected from the Second Affiliated Hospital of Army Medical University through the process as follow. The inclusion criteria for the prospective validation study were participants over 18 years old who need to underwent CXR examination written out by clinicians. Before the participants underwent the examination in the radiology department of our hospital, we fully informed them the content of the prospective study. Only after they agreed to participate in the study and signed the informed consent form, their CXRs will be collected. From Oct 1 to Oct 15, 2023, 1517 participants signed 1540 copies of the informed consent forms (a few participants underwent two or three times CXR examinations) and participated in this study. A total of 1540 posteroanterior CXRs and 1161 lateral CXRs were collected. After excluded the lateral CXRs, the remaining posteroanterior CXRs were constructed as the prospective test dataset.

The manufacturer and device information of all the above CXRs is displayed in Supplementary Table 1. The general information and filtering details of the held-out, multicentre, recombination and prospective test datasets are shown in Supplementary Table 4.

## Annotations for categories and localizations

Our annotations included both category labels and localizations with bounding boxes for each abnormality. We focused on 14 common abnormalities in CXRs, including atelectasis, calcification, consolidation, effusion, emphysema, fibrosis, fracture, mass, nodule, pleural thickening, pneumatosis, pneumothorax, postoperative metal and venipuncture. The implication of each abnormality can be found in Supplementary Table 25.

The standardization of the annotations is extremely important to ensure annotations with high quality[29]. To ensure the accuracy of the GT annotations, we formulated three general rules for the annotation of each bounding box by referring to the labelling principles of target detection. (1) The bounding box should be the minimum external rectangle box that contains the whole abnormality in principle. It is worth noting that for the annotation of small abnormalities, such as tiny nodules and pacemaker tip electrodes, the minimum external rectangular box can be appropriately enlarged. (2) If the edge of one abnormality is obscured or sheltered, the edge should be confirmed according to the knowledge and experience of radiologists. For example, when the lower boundary of the pleural effusion is difficult to determine, radiologists need to use their knowledge and experience to determine the lower boundary of the pleural effusion according to the position of the contralateral hemidiaphragm and costophrenic angle, as well as further annotate the effusion (Supplementary Fig. 15). (3) If there are multiple abnormalities with the same category in the same CXR, each abnormality should be individually annotated by a bounding box as long as they are not connected. Under the above general rules, the details of the annotation principles for each abnormality are described in Supplementary Table 25, and annotation examples for each abnormality are given in Supplementary Fig. 16. All the radiologists were trained according to the details of annotation principles for each abnormality and made full sense of them before participating in the annotation process.

Both the category and localization of each abnormality in CXRs were annotated by the LabelImg tool (v1.8.0, https://pypi.org/project/labelImg/1.8.0/).

## Construction of the CXR-AL14 dataset via the human-in-the-loop approach

Accurately annotating multiple abnormalities on a large number of CXRs is a massive challenge. To address this issue, we employed an approach named human-in-the-loop in which humans and models work in tandem (Fig. 9); this is similar to the labelling process in Greenwald's study[30]. First, six expert radiologists (with more than 20 years of experience in radiological diagnosis) collectively annotated all abnormalities on 8,000 CXRs, which were adopted to train a preliminary deep model. During this process, the six experts jointly formulated the GT bounding boxes for each CXR, and if there were any conflicts of opinion, they discussed together to obtain the final decision. Second, predictive annotations for the new unannotated CXRs were generated by the preliminary model, and these annotations were randomly sent to twelve senior radiologists (with more than 10 years and less than 20 years of experience in radiological diagnosis) to correct the categories, sizes and localizations of the bounding boxes. The corrected annotations were then randomly dispensed to the six expert radiologists for further correction. The checks by two levels of radiologists are analogous to the routine workflow of radiologists interpreting CXRs. In this process, if one expert had doubts about any annotation, a final decision was made after discussion with another expert radiologist. After expert correction, the CXR annotations could be considered GT-level annotations. Then, the CXRs with corrected annotations were added to the training dataset for retraining the model and sent to the final dataset. As the number of iterations increased, the predictive performance of the model increased, and the correction workload of the radiologists

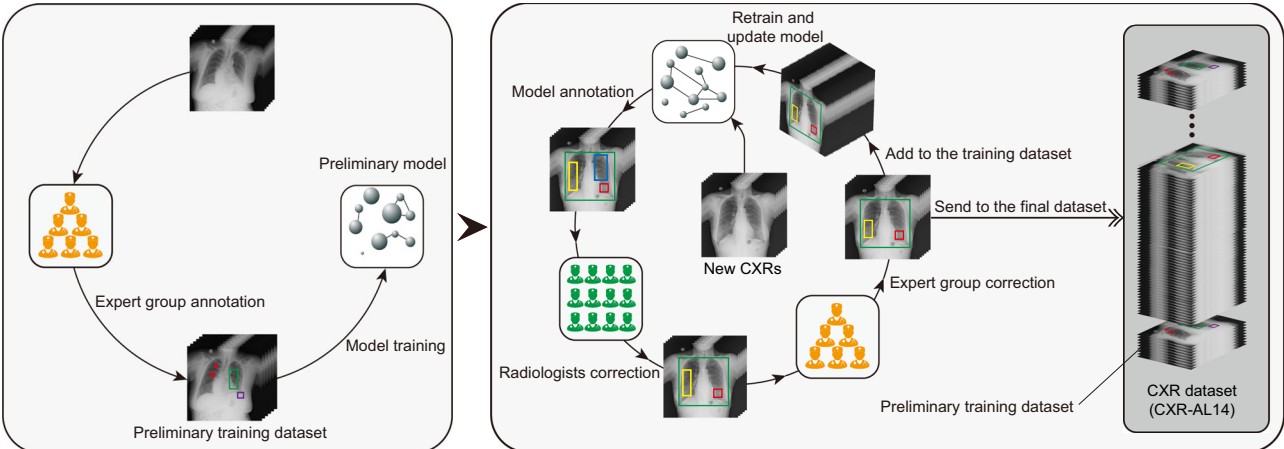

**Fig. 9 | The creation process of the CXR-AL14 dataset via the human-in-the-loop approach.** In the first stage, six expert radiologists (with more than 20 years of experience) collectively annotated the GT bounding boxes of all abnormalities on 8,000 CXRs, which were adopted to train a preliminary deep model and sent to CXR-AL14 dataset. In the second stage, predictive annotations for the new unannotated CXRs were generated by the preliminary model, and these annotations were randomly sent to twelve senior radiologists (with 10-20 years of experience) to correct the categories, sizes and localizations of the predicted bounding boxes.

After first correction, the corrected annotations were then randomly dispensed to the six expert radiologists for second correction. The checks by two levels of radiologists are analogous to the routine workflow of radiologists interpreting CXRs. After expert correction, the annotations of CXR could be considered GT-level annotations. Then, the CXRs with corrected annotations were added to the training dataset for retraining the model and sent to the final dataset. After seven iterations, the CXR-AL14 dataset was created. GT: ground-truth.

gradually decreased. After seven iterations, a large CXR dataset named CXR-AL14 was created, which contained 165,988 CXRs (102,904 abnormal CXRs and 63,084 CXRs with "No finding" labels) from 144,968 patients with 253,844 GT categorical and localization bounding boxes. There were 47,016 CXRs with one abnormality, 28,683 with two abnormalities, and 27,205 with multiple abnormalities in the CXR-AL14 dataset; more details can be found in Supplementary Table 26. The volume of CXRs for each iteration, and the number of each abnormality annotated by each expert are shown in Supplementary Tables 27-28. Moreover, the performance of the updated model after each iteration was evaluated on the held-out test dataset.

To ensure the accuracy and reliability of the GT labels in the CXR-AL14 dataset, the intrareader and intrareader variability of the annotation of six experts in the expert group was evaluated. Since the six experts played the final role in the creation of the CXR-AL14 dataset, we only observed the interreader and intrareader variability among these six experts. First, 600 CXRs were randomly selected from the CXR-AL14 dataset (excluding the 8000 CXRs for training the preliminary model) and interpreted by the preliminary model. Then, each radiologist of the twelve radiologists was given 50 CXRs on average for correction (the process is completely consistent with the human-in-the-loop approach). Finally, we sent the 600 CXRs with corrected annotations to six experts for further correction. After each expert independently corrected all 600 CXRs for the first time, the IoU values of each abnormality between pairs of experts were calculated to observe the interreader variability[31]. One month later, the annotations of all 600 CXRs corrected by the twelve radiologists were independently corrected by the six experts for a second time to evaluate the intrareader variability by calculating the IoU values of each abnormality between two annotations.

### Development of the YOLOX model
In the study, the YOLOX architecture[32] was used to train the required deep model. We randomly divided all CXRs from the CXR-AL14 dataset into training and tuning datasets at a ratio of 9:1 (Table 1). The training dataset was used to train the YOLOX model for the identification and localization of 14 abnormalities, and the hyperparameters of the YOLOX model were adjusted by the tuning dataset during the training

process. Note that the YOLOX architecture was used to train the required deep model; the backbone of this network is the cross-stage partial (CSP) Darknet, which mainly includes a focus stem and a CSP layer. Its patch aggregation-based feature pyramid structure integrates underlying and superficial features, and the design of its decoupled head contributes to the convergence of the model (Supplementary Fig. 17). As a high-performance anchor-free detector in the YOLO series, YOLOX achieves a balance between speed and accuracy. During the training and tuning stages, the size of each input image was 1280×1280, the batch size was 8, and the number of training epochs was 30. Note that data enhancement strategies such as mosaicking, mix-up, random flipping and colour jitter were also adopted, the stochastic gradient descent (SGD) strategy was selected as the model optimizer, and the initial learning rate was set to 0.01 with a cosine schedule. Furthermore, we also used a pretrained model on the Microsoft Common Objects in Context (MS COCO) dataset to improve the training efficiency of YOLOX. The epoch value was an important hyperparameter for the training of the YOLOX network. An epoch value that is too small could lead to underfitting of the model, while an epoch value that is too large could lead to overfitting of the model. The relationship between the loss values and the epoch values during the training of the YOLOX model is given in Supplementary Fig. 18.

In addition, the other two localization networks, including Faster R-CNN[33] and RetinaNet[34], were further separately trained on the CXR-AL14 dataset. In this study, YOLOX, Faster R-CNN and RetinaNet were all implemented with PyTorch 1.10.1. All training, validation, and test procedures were conducted on eight NVIDIA RTX 2080Ti graphics processing units (GPUs).

### Performance evaluation of the YOLOX model
Five validations were performed in our study. First, 5-fold cross-validation was performed on the CXR-AL14 dataset, and the five models were further tested on the held-out test dataset. Second, the held-out test dataset was used to evaluate the general performance of the YOLOX model. In addition, the Faster R-CNN model and RetinaNet model were also tested on the held-out test dataset, and the performance of these two models was compared with that of the YOLOX model. Third, to assess the generalization of the YOLOX model, a multicentre validation was performed by using CXRs from four

participating hospitals. Moreover, the IoU was used as a final evaluation indicator to demonstrate the localization accuracy achieved by the YOLOX model. On the premise of correctly identifying categories, we further calculated the average IoU value of the predicted bounding boxes relative to the GT bounding boxes for each abnormality on the external test dataset (all CXRs come from four multicentres). Fourth, using the held-out test dataset and the recombination dataset, the performance of the YOLOX model was compared with that of six radiologists, including three senior radiologists (with 15-18 years of experience) and three junior radiologists (with 6-8 years of experience) who did not participate in the creation of the CXR-AL14 dataset. Fifth, a prospective validation was conducted on the 1,540 CXRs to test the clinical applicability of the YOLOX model. After the YOLOX model was developed, we simulated clinical practice and used this model to interpret the newly generated CXRs. Then, two expert radiologists annotated the GT bounding boxes for these CXRs without knowing the results achieved by the YOLOX model. Finally, the prospective validation results could be obtained for the YOLOX model by comparing the predicted bounding boxes with the GT bounding boxes.

In order to accurately assess the model performance, all the GT bounding boxes for the CXRs from the held-out test dataset, the multicentre test datasets A-D, recombination test dataset and prospective test dataset were manually annotated by two expert radiologists from the expert group together without the help of the human-in-the-loop approach. If the two expert radiologists disagreed on the annotation of any abnormality, a third expert radiologist was required to reach a final conclusion. The abnormality distribution in the above CXRs is shown in Supplementary Table 4.

## Stress test for the YOLOX model

A quality control stress test for the YOLOX model was performed on the held-out test dataset. The quality of CXRs is primarily exhibited by their brightness and contrast. Thus, we tested the performance of the YOLOX model with various degrees of brightness and contrast changes in CXRs. Negative and positive ranges of both brightness and contrast were tested, respectively, with increments or decrements of 0.1, where 1.0 was the baseline.

## Development and validation of the automatic CTR calculation algorithm

A total of 12,000 CXRs randomly selected from the included CXRs were applied to develop and validate an automatic CTR calculation algorithm. Ten thousand CXRs selected randomly from the 12,000 CXRs were used to manually generate the GT masks of both lungs and the heart by two junior radiologists. Note that if both radiologists were unable to effectively segment both lungs or the heart from the same CXR due to the edge of the heart being obscured by some abnormalities, this CXR was excluded. If only one junior radiologist could not segment them, a senior radiologist would be invited to jointly decide whether to continue the segmentation process or exclude this CXR. Finally, 426 CXRs were excluded, and the remaining 9,574 CXRs with GT masks (at a ratio of 9:1 for training and fine-tuning) were used to develop a deep model for segmenting both lungs and the heart in CXRs based on the attention UNet architecture[35]. Once the segmentation mask was obtained for each CXR, the CTR could be calculated automatically by using the proposed algorithm. (Supplementary Fig. 19)

The CTR can be calculated using the convention of measuring the thoracic diameter (L) as the distance from the inner margin of the ribs at the level of the dome of the right hemidiaphragm and the cardiac diameter ($L_1 + L_2$) as the horizontal distance between the rightmost and leftmost margins of the cardiac shadow. In our study, the calculation of the CTR was based on the segmentation of both lungs and the heart in CXRs. In particular, we employed attention UNet to develop the deep model for the segmentation of both lungs and the heart, where the input image size was $512 \times 512$, the number of epochs was set to 100

and the batch size was set to 4. The initial learning rate was 0.00001; when the loss of the validation set increased for two consecutive epochs, the learning rate was reduced by half. Once the binary mask of both lungs and the heart was obtained, the thoracic diameter (L) and the cardiac diameter ($L_1 + L_2$) could be calculated by the proposed algorithm, where the process for doing so is described as follows.

Step 1: The closed morphology operation was applied to the mask to eliminate holes and burrs.

Step 2: By using the connected area labelling approach, the two largest connected areas of the lung and the largest connected area of the heart were preserved in the mask.

Step 3: The Canny operator was applied to the mask to obtain the single-pixel boundary between the lung and the heart.

Step 4: Traversing upwards from the lowest point of the right lung boundary using a number of horizontal lines, at least two intersections were encountered between each horizontal line and the boundary. The first two intersections were determined from left to right, and the distance between these two intersections was calculated. As the upwards traversal process proceeded, a number of difference values between two adjacent distances could be obtained. Note that the horizontal line that determined the maximum difference value can be regarded as a tangent line passing through the right diaphragm.

Step 5: The width of the chest (L) could be calculated by the two farthest intersections determined by the above tangent line and the boundary of both lungs.

Step 6: Find the left and right points that were farthest from the midline on the boundary of the heart, and the distances from the above two points to the midline were calculated and summed ($L_1 + L_2$).

Step 7: The CTR could be easily calculated according to the formula ($L_1 + L_2$)/L.

The other 2,000 CXRs were used to evaluate the accuracy of the CTR calculation algorithm. The above two junior radiologists were also invited to manually calculate the CTR of each CXR according to the clinical measurement rules, and their average values were considered as the reference standards. If both radiologists were unable to calculate the CTR, the corresponding CXR was excluded. If only one junior radiologist could not calculate it or the difference between the CTR results measured by two junior radiologists was greater than 0.05, the above senior radiologist was invited to achieve a final consensus result. Ultimately, 1950 CXRs were included when constructing a test dataset.

In addition, 500 CXRs randomly selected from each hospital were used for multicentre validation, and 300 randomly selected CXRs were used for prospective validation. According to the above CTR screening rule, the final numbers of included CXRs were 498, 495, 484, 492 and 293, respectively. The detailed manufacturer and general information of all the above CXRs can be seen in Supplementary Tables 23 and 24.

## Statistical analysis

The general information of CXRs in the CXR-AL14 dataset was conducted using numpy package (version 1.19.3) in Python (version 3.7.9). We employed the precision, recall, AP, mAP and F1-score to evaluate the performance of the YOLOX model. The performance of the YOLOX model was evaluated using the above metrics with different IoU thresholds, which are indicators for measuring the degree of overlap between the predicted bounding box and the GT box; the larger the IoU threshold is, the higher the degree of overlap. For each abnormality, the precision-recall curve could be easily drawn, and the AP value could be obtained by calculating the area under the precision-recall curve. Furthermore, the mAP could be calculated by averaging the AP values of all abnormalities. The McNemar test and chi-square test were carried out using the statsmodels (version 0.13.2) and scipy (version 1.5.4) packages in Python (version 3.7.9), and the 95% Wilson confidence interval was applied to both the precision and recall. ANOVA and the LSD method for pairwise comparisons were conducted using SPSS Statistics (Version 22.0.0, IBM SPSS Statistics). Bland–Altman

analysis and the ICC were employed to measure the precision of the CTR value calculated by the proposed framework. Note that a *P* value less than 0.05 was considered statistically significant.

## Reporting summary

Further information on research design is available in the Nature Portfolio Reporting Summary linked to this article.

## Data availability

All data supporting the findings described in this manuscript are available in the article and in the Supplementary Information. According to relevant national regulations, there are certain restrictions on the number of medical images for publicly available. Therefore, the CXR-AL14 dataset is partially available for public use (nearly 100,000 CXRs), interested researchers can contact the corresponding author Dong Zhang (hszhangd@tmmu.edu.cn.) or visit this website [cxr-al14.top] to request access. In addition, the total CXR-AL14 dataset is available for online use by requested on the website [https://www.ncmi.cn//phda/dataDetails.do?id=CSTR:17970.11.A0048.202312.605.V1.0]. It should be noted that the CXR-AL14 dataset will only be available for academic research, and not for other purposes. Interested researchers need to register their personal and institutional information on above websites and send data access requests to the web administrator. The web administrator and corresponding author will review the requests for consideration and respond within two weeks. Once approved, the dataset can be used by the interested researchers. Note that interested researchers who have utilized the CXR-AL14 dataset for research must cite this article.

The multicentre test datasets are not available for public use. If the interested researchers want to achieve the multicentre test datasets for non-commercial use, they can request for the corresponding author Dong Zhang (hszhangd@tmmu.edu.cn.). Corresponding author will review their requests and ask for consent from each centre, requestors will receive a response within two weeks. Source data are provided with this paper.

## Code availability

The program codes of the YOLOX model and the CTR calculation algorithm in this study are publicly available, which can be downloaded at https://github.com/CXR-AL14/CXR-Code. DOI link: https://doi.org/10.5281/zenodo.8120660.

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

## Acknowledgements

This work was supported by the clinical major innovative characteristic technology project of Second Affiliated Hospital of Army Medical University (2018JSLC0016 for D.Z.) and by the Talent project of Chongqing (CQYC202103075 for D.Z.). We thank the information department in Second Affiliated Hospital of Army Medical University for providing extensive CXRs data and all those who had made contribution to the publication of the work.

## Author contributions

D.Z. and W.J.F. participated in the conception, the design. J.Q. and Y.J.N. participated coordination of the study. W.J.F. and Y.Y. participated manuscript preparing. S.W. participated in the CXR collection of CXR-AL14 dataset. G.X.W., Y.X., X.T.F. and Q.H.W. provided CXR data for multicentre validation. D.Z., L.W., C.W.L., L.Y., Y.L. and Q.C.Z. constitute members of the expert radiologist group for establishing the GT boxes, and participate in the formulation of GT bounding boxes. Jia.C., Y.Q.C., Y.Z., L.Q.Y., H.G., S.S.Z., G.Z., X.Z., M.X.X., H.L.X., W.J.F. and Jiao.C. participated in the correction of multiple abnormalities of CXR by the human-in-the-loop approach. L.Z., B.W., G.F.L., X.D.G., S.H. and J.L. participated in the human-model comparison on the held-out and recombination test datasets. S.L.X., K.M. and T.W.X. participated in the calculate of CTR on CXRs. J.Q., X.C.C., M.H., J.J.X. and Y.Y. participated in data analysis and model establishment and optimization. All authors read and approved the final manuscript.

## Competing interests

The authors declare no competing interests.
