## [Peer Review File · Nature Communications]

A deep-learning-based framework for identifying and localizing multiple abnormalities and assessing cardiomegaly in Chest X-rayREVIEWER COMMENTS

Reviewer #1 (Remarks to the Author):

The authors proposed a new dataset of thoracic abnormalities in chest X-rays with over 165k images and over 250k bounding boxes (of the abnormalities). The bounding boxes are generated using a pre-trained model and verified/corrected by at least one radiologist and one expert. Then, a YOLOX model is trained using the composed dataset and testing on the left-out testing set and external validation sets. Superior results of the proposed method are reported in comparison to a group of human radiologists (three junior and three senior) on the held-out testing set. The main contribution lies in composing the dataset. But, there are flaws during the dataset-building process, especially in the annotation part. The overall presentation is easy to follow. Though, I have many critical concerns that are detailed below:

-I have critical concerns about how the annotation GTs were generated for the testing and validation data. The details of how the annotations are conducted are missing, e.g., How many "experts" are employed for annotating each data sample? What does "over 10 years' radiological experience" mean? How are the conflicts (among different experts) resolved? How are the data samples distributed if only one expert annotates one data sample? What's the distribution of experience among those experts? How could that affect the performance evaluation?

- Following the above comment, I am concerned about the reported superior performance of the trained YOLOX system compared to the human radiologists, especially those senior ones. Are those senior radiologists more senior than those experts employed for the annotation tasks? Which one is the better one, testing GTs or testing results? The unclarity of the annotation process makes the reported results less convincing.

- Also, the superior results are reported on the held-out testing set (whose GTs are first generated by the algorithm and verified by the radiologists). I am not surprised that the algorithm-generated results are stabler and better aligned with the GTs. It will be valuable to compute the inter-observer variance (including the algorithm) to show the difference across different (grouped) radiologists.

- It will also be helpful to validate the accuracy of the bounding box annotations in the training set. Even though it may not affect the final performance since the model training may suppress the effects of some of the annotation errors.

- It will also be helpful to analyze the results of the external validation set in detail. For example, how the expert group will affect the performance. How are the external data varied from the training data (domain gaps), e.g., on each abnormality category?

- The intention of adopting YOLOX is questionable instead to other object detection frameworks since the series of YOLO detection methods are more outstanding in the real-time object detection task. However, its accuracy is primarily equivalent and often worse than the anchor-based/region-proposal-based methods, e.g., Faster-RCNN, etc. It is unclear about the design choices here.

Reviewer #2 (Remarks to the Author):

The authors make a keen observation that AI/ML-based diagnostic methods for chest x-rays tend to focus on a single pathology when more than one might exist. They also observe that sufficient training power is needed for every pathology to be able to accurately detect and localize the disease toward aiding the radiologist. They observe that heat maps are insufficiently accurate as a localization method.

To respond to the above, the authors built a large chest x-ray dataset with 14 pathologies. They trained a YOLOX localizer and detector for them. The performance of the detector is quite good, however, they have had to compromise on the IoU-T (threshold) values such that only 10% IoU is considered acceptable. Nonetheless, it is a move in the right direction.

However, the manuscript leave much unsaid and in my opinion is not ready for publication.

Criticisms/corrections/comments:

1. The structure of the manuscript is really weird. The authors are recommended to restructure content (and trim text) following a more standard structure

Introduction, Prior Art, Data, Methods, Experiments, Analysis, (Other contributions), Conclusions, References

There is high redundancy in the material and significant cuts / edits to the text will still leave the impact in place.

2. Data set development - what was the inter-reader and intra-reader variability following the human-in-the-loop approach. What was the performance of the algorithm? How many iterations? How many images were reviewed per iteration? How many images per disease per reviewer? What is the overlap in the diseases per image, i.e., how many images have multiple diseases?

3. What is the performance of the algorithm as a function of the variability inherent in the dataset?

4. Why is the algorithm performing poorly for two diseases?

5. Is it important to describe the tool that has been developed? If the tool must be a part of the manuscript, what is its performance in active use? What approximations / engineering was done to ensure quality control for the tool?

Reviewer #3 (Remarks to the Author):

The aim of this paper is two folds for identifying and localizing 14 abnormalities and calculating the cardiothoracic ratio simultaneously with about 165 thousand CXRs and 253 thousand bounding boxes. This is quite interesting and clinical relevant topics. However, there are several concerns on a lack of technical novelty and the reproducibility of this study. I recommend this paper would be better for radiology related clinical journals.

1. Is this dataset (CXR-AL14) collected in consecutive manner? How do you differentiate AP CXR vs PA CXR? In addition, there is a lack of details (selection and exclusion criteria, duration, consecutive or not, etc) on dataset of external and prospective validations.

2. There is no external or cross-validation for evaluating the reproducibility of this method. Especially, leave-one-out methods could predict over-estimation. Please use 3-5 fold cross-validation or external validation

3. In case of annotation rule 2 (if the edge of one abnormality is obscured or ...) or especially in case of disagreement among radiologists, how to confirm according to the knowledge and experience of radiologists. Describe in detail.
4. In case of human in the loop approach, please describe the accuracy improvements in each iteration.
5. There is a lack of methodological originality.
6. There is no stress test, ablation and parameter changing studies for evaluating optimization of models.

Reviewer #4 (Remarks to the Author):

This study developed and validated a YOLOX model for identifying and localizing 14 abnormalities and assessing cardiomegaly in Chest X-ray with a large-scale datasets containing 165,988 radiographs and 253844 bounding boxes.

1. Bounding box segmentation has some difficulty for labeling pulmonary lesions on chest radiograph because a box cannot follow anatomical structures or lesion shape. Pixel-based annotation is usually adopted for commercially available AI-based diagnosis-software for chest radiographs. For example, in other study (reference #2), they developed a high-performance model with 146717 chest radiographs with pixel-based annotation for 10 abnormalities.
2. IOU-threshold is a usual method to evaluate the accuracy of YOLOX model. However, qualitative evaluation of lesion detection on chest radiograph by expert radiologists might be more accurate method to evaluate the performance of the model. It could be done on external validation process.
3. False positive rate (the number of false positive findings per image) is an important index to see how much attention and effort would be additionally needed to differentiate false positive lesions to read chest radiographs in workflow if the software is adopted in clinical reading session.
4. Comparison of performance of the YOLOX model with the radiologists better be conducted in external test sets as well. With the validation dataset from the one hospital which is the same source of training dataset may have some performance gain to YOLOX model than radiologists.

5. There are already large-scale open-source datasets of chest radiograph up to 224316 chest radiographs (MIMIC-CXR) and 174 radiologic findings (PADChest). All 165988 chest radiographs with 14 abnormalities represented by box segmentation constructed in this study should be another good open-source dataset if it is opened to public. Did the investigators develop this dataset for public use?

6. Development of the X-ray Eye software in the manuscript is not appropriate because it is not appropriately described in purpose, method, and result with appropriate hypothesis, validation, and results for automatic transmission of images from the PACS to the software and priority prompt given for pneumothorax or pneumatosis, which could significantly reduce the time to obtain reported results according to the assertion of the authors.

7. In discussion section, It would better describe detailed comparison of performance and what is incremental advance of this study results upon previous studies which has been published already with large-scale datasets.

8. Prospective validation of the performance of the YOLOX model should be prospective fashion regarding methodology and should be described in more detail.

9. In CTR measurement, the maximal distance from the inner margin of the ribs at any level is commonly conducted in clinical practice not at the level of the dome of the right hemidiaphragm as used in this study.

10. Chest radiograph is routinely obtained with posterior anterior projection (PA), but in intensive care unit or for severely ill patients anterior-posterior projection (AP) is usually used to get chest radiograph. CTR cannot be accurately measured in Chest AP.

RESPONSE TO REVIEWER COMMENTS

Please note that the original comments are in a red font, and our responses are in a black font.

Reviewer #1 (Remarks to the Author):

The authors proposed a new dataset of thoracic abnormalities in chest X-rays with over 165k images and over 250k bounding boxes (of the abnormalities). The bounding boxes are generated using a pre-trained model and verified/corrected by at least one radiologist and one expert. Then, a YOLOX model is trained using the composed dataset and testing on the left-out testing set and external validation sets. Superior results of the proposed method are reported in comparison to a group of human radiologists (three junior and three senior) on the held-out testing set. The main contribution lies in composing the dataset. But, there are flaws during the dataset-building process, especially in the annotation part. The overall presentation is easy to follow. Though, I have many critical concerns that are detailed below:

Response: Thank you for your valuable comments and encouragement. We apologize that the details of the annotation process were not described well in the original manuscript. In the revised version, the annotation process has been described in detail. We also addressed your critical concerns in detail. We believe that the manuscript has been markedly improved after our revisions based on your and the other reviewer comments. Thank you again for your suggestions.

-I have critical concerns about how the annotation GTs were generated for the testing and validation data. The details of how the annotations are conducted are missing, e.g., How many "experts" are employed for annotating each data sample? What does "over 10 years' radiological experience" mean? How are the conflicts (among different experts) resolved? How are the data samples distributed if only one expert annotates one data sample? What's the distribution of experience among those experts? How could that affect the performance evaluation?

Response: We appreciate your comments about how the GT annotations were generated.

The annotation quality of the dataset directly determines whether the developed model can obtain good performance. We apologize that the annotation process was not described well in the original manuscript. We have described the annotations in more details in **Part 4.4** and **Part 4.6** (the yellow highlighted text) in the revised manuscript, as follows:

To ensure accurate GT labels, we performed the following three tasks before creating the CXR-AL14 dataset. First, we formulated three general rules for the annotation of the bounding boxes according to the annotation principles of object detection, which were originally described in **Part 4.3**. Second, all the radiologists involved in the creation of the CXR-AL14 dataset were trained according to the annotation details and principles for each abnormality, which was originally mentioned in *Supplementary Table 23*. Only after the radiologists made full sense of the annotation principles and method, they could take part in the creation

of the CXR-AL14 dataset. Third, the 8000 CXRs used to train the preliminary model were annotated by the expert group through collective discussion, which ensures the good performance of the preliminary model.

In addition, in order to accurately evaluate the model performance, the GT bounding boxes of all test datasets, including the held-out test dataset, multicentre test datasets A-D, recombination test dataset (700 CXRs were randomly selected from each multicentre, with a total of 2,800 CXRs in this dataset) and prospective test dataset were not generated by the human-in-the-loop approach but were manually annotated by two experts from the expert group together, which was mentioned in **Part 4.6**.

In our study, the validation dataset was called the tuning dataset which was randomly selected from the CXR-AL14 dataset. Thus, the generation of GTs in validation dataset was consistent with that in the CXR-AL14 dataset.

-How many "experts" are employed for annotating each data sample?

Response: In this study, at least two radiologists, including one expert with more than 20 years of experience and one radiologist with more than 10 years of experience in radiological diagnosis, interpreted each CXR during the creation of the CXR-AL14 dataset. Detailed information on the experts and radiologists that annotated each dataset is shown in **Table R1-1**. We added this content to the corresponding part of the revision manuscript.

Table R1-1 Details of the experts involved in the annotation of each example in each dataset

Dataset	Experts or radiologists involved in the annotation
CXR-AL14 dataset (8000 CXRs for training the preliminary model)	Each CXR was annotated by the expert group (six experts) after collective discussion, ensuring that the preliminary model showed reliable performance.
CXR-AL14 dataset (Except for the above 8000 CXRs)	The annotations for each CXR were generated via the human-in-the-loop approach. First, the annotations were corrected by one radiologist with more than 10 years of experience in radiological diagnosis. Then, the annotations were reviewed and corrected by an expert radiologist. The checks by two levels of radiologists are analogous to the routine workflow of radiologists interpreting CXRs. If one expert had doubts about any annotation, a final decision was made after discussion with another expert radiologist.
Held-out test dataset, multicentre, recombination and prospective test datasets	Each CXR was manually annotated by two experts from the expert group together without the help of the human-in-the-loop approach. If the two experts disagreed on the annotation of any abnormality, the results were discussed with a third expert to reach a final conclusion.

-What does "over 10 years' radiological experience" mean?

Response: Thank you for your good comments. We apologize for the unclear expression. "Over 10 years' radiological experience" means that the radiologist had more than 10 years of work experience in radiological diagnosis. We have revised this phrase to read "more than 10 years of experience in radiological diagnosis". (P18, Line 13)

(Continued on the next page)

-How are the conflicts (among different experts) resolved?

Response: The relevant details are shown in **Table R1-1** above. These details can be summarized as follows:

For the 8000 CXRs that were used to train the preliminary model, the expert group resolved the conflicts through discussion.

For the CXR-AL14 dataset (except 8000 CXRs for training preliminary model), The annotations for each CXR were generated via the human-in-the-loop approach. First, the annotations were corrected by one radiologist with more than 10 years of experience in radiological diagnosis. Then, the annotations were reviewed and corrected by an expert radiologist. The checks by two levels of radiologists are analogous to the routine workflow of radiologists interpreting CXRs. If one expert had doubts about any annotation, a final decision was made after discussion with another expert radiologist.

For all test datasets, including the held-out, multicentre, recombination and prospective test datasets, each CXR was manually annotated by two experts from expert group together without the help of the human-in-the-loop approach. If the two experts disagreed on the annotation of any abnormality, the results were discussed with a third expert to reach a final conclusion.

-How are the data samples distributed if only one expert annotates one data sample?

Response: We counted the number of abnormalities annotated by each expert in the creation of the CXR-AL14 dataset. Each expert annotated more than 37,000 bounding boxes. The detailed information is shown in **Table R1-2** which has been added to the Supplementary Materials (*Supplementary Table 26*). The abnormality distribution of the 8000 CXRs annotated by the expert group to train the preliminary model is also shown in the table.

Table R1-2 Number of CXRs and each abnormality annotated by each expert.

	Expert group	Expert 1	Expert 2	Expert 3	Expert 4	Expert 5	Expert 6	Total
CXRs	8000	26329	26329	26332	26332	26332	26334	165988
Atelectasis	17	61	61	57	52	36	74	358
Calcification	1640	6278	5471	5254	5441	5015	4255	33354
Consolidation	1002	3533	3281	3135	2921	2797	4597	21266
Effusion	2015	6847	6488	6854	6421	6519	8967	44111
Emphysema	1159	3779	3966	4057	4027	3718	2884	23590
Fibrosis	750	2835	2486	2452	2490	2301	3373	16687
Fracture	575	2328	1927	1907	1793	1564	2538	12632
Mass	132	469	436	416	394	441	710	2998
Nodule	2170	8724	6518	6620	7771	7131	7043	45977
Pleural thickening	336	1685	1084	1218	962	872	1236	7393
Pneumatoxis	259	461	739	839	1068	1125	2462	6953
Pneumothorax	313	715	941	910	1002	1094	2089	7064
Postoperative Metal	866	3679	2629	2644	2884	2970	4257	19929
Venipuncture	482	894	1230	1472	1784	2004	3666	11532
Total (Annotations)	11716	42288	37257	37835	39010	37587	48151	253844

-What's the distribution of experience among those experts? How could that affect the performance evaluation?

Response: All six radiologists in the expert group had more than 20 years of working experience in radiological diagnosis. In the radiology department of the hospital, the radiologists are not divided into subgroups in their daily clinical practice. Thus, all six experts had extensive experience with CXR interpretation. In addition, expert radiologists with more than 20 years of experience in radiological diagnosis are fully capable of interpreting abnormalities on CXRs. We also evaluated the intervariability and intravariability of the six experts' CXR annotations. The average intrareader IoU value of each abnormality ranged from 0.773 to 0.992, and the average interreader IoU value of each abnormality ranged from 0.698 to 0.968. More details are shown in **Table R1-3**. Furthermore, we counted the number of bounding boxes with IoU=0 for each abnormality and found that the maximal number was 14 in the interreader study (<0.023 per CXR) and the maximal number was 6 in the intrareader study (<0.01 per CXR). More details are shown in **Table R1-4**. These results demonstrated that the experts' annotations of CXRs have good consistency (interreader and intrareader). In addition, the GT annotations of all test datasets were manually annotated by two experts together. If two experts disagreed with the annotation of the identical abnormality, a third expert was invited to offer their opinion to reach a final decision. Therefore, we believe that the experience of the experts is sufficient to obtain a correct diagnoses and annotations of each CXR, and our approach has little impact on the performance of the model.

(Continued on the next page)

Table R1-4 Number of bounding boxes with IoU=0 for each abnormality on interreader or intrareader variability

	Atelectasis	Calcification	Consolidation	Effusion	Emphysema	Fibrosis	Fracture	Mass	Nodule	Pleural thickening	Pneumato- sis	Pneumoth- orax	Postoperat- ive metal	Venipunct- ure
Inter- reader (N=600)	E 1 vs E 2	0	1	2	0	0	2	0	0	3	0	0	0	0
	E 1 vs E 3	0	7	0	0	0	7	0	0	13	0	0	0	0
	E 1 vs E 4	0	2	2	0	0	2	0	0	10	0	0	0	0
	E 1 vs E 5	0	2	6	0	0	2	0	0	1	0	0	0	0
	E 1 vs E 6	0	4	3	0	0	2	0	0	11	0	0	0	0
	E 2 vs E 3	0	8	2	1	0	5	0	0	9	0	0	0	0
	E 2 vs E 4	0	0	0	0	0	0	0	0	7	0	0	0	0
	E 2 vs E 5	0	0	4	0	0	0	0	0	3	0	0	0	0
	E 2 vs E 6	0	2	1	0	0	0	0	0	9	0	0	0	0
	E 3 vs E 4	0	9	2	1	0	5	0	0	14	0	0	0	0
	E 3 vs E 5	0	9	6	1	0	5	0	0	12	0	0	0	0
	E 3 vs E 6	0	7	3	1	0	5	0	0	10	0	0	0	0
	E 4 vs E 5	0	0	4	0	0	0	0	0	10	0	0	0	0
	E 4 vs E 6	0	2	1	0	0	0	0	0	10	0	0	0	0
	E 5 vs E 6	0	2	5	0	0	0	0	0	10	0	0	0	0
Intra- reader (N=600)	E 1 vs E 1*	0	1	1	1	0	0	1	0	2	0	0	0	0
	E 2 vs E 2*	0	1	0	0	0	2	0	0	2	0	0	0	0
	E 3 vs E 3*	0	0	1	0	0	2	0	0	2	0	1	0	0
	E 4 vs E 4*	0	1	2	0	0	1	0	0	1	0	0	0	0
	E 5 vs E 5*	0	1	1	0	0	4	1	0	6	1	0	0	0
	E 6 vs E 6*	0	0	1	0	0	1	1	0	1	0	0	0	0

*. The twice annotations of each expert.

- Following the above comment, I am concerned about the reported superior performance of the trained YOLOX system compared to the human radiologists, especially those senior ones. Are those senior radiologists more senior than those experts employed for the annotation tasks? Which one is the better one, testing GTs or testing results? The unclarity of the annotation process makes the reported results less convincing.

Response: Thank you for your valuable question.

In the performance comparison between the YOLOX model and radiologists, we employed two levels of radiologists, including three senior radiologists and three junior radiologists, who did not participate in the creation of the CXR-AL14 dataset. These senior radiologists were different from those experts in the expert group. First, their working experience was less extensive than that of the experts; thus, the experts who participated in annotating the CXR-AL14 dataset had more experience than these three senior radiologists. Second, these senior radiologists were not involved in the creation process of the CXR-AL14 dataset.

The bounding boxes that were manually annotated by the expert group were the GT bounding boxes. We compared the results of the senior radiologists, junior radiologists and YOLOX model with the GTs. Then, the performance of the YOLOX model was compared with that of the junior and senior radiologists to observe which level of radiologists the model could reach. We added this information to the revised manuscript.

- Also, the superior results are reported on the held-out testing set (whose GTs are first generated by the algorithm and verified by the radiologists). I am not surprised that the algorithm-generated results are stabler and better aligned with the GTs. It will be valuable to compute the inter-observer variance (including the algorithm) to show the difference across different (grouped) radiologists.

Response: Thank you for your helpful comment.

As you noted, if the GT labels of the held-out test dataset were annotated by the human-in-the-loop approach and corrected by the radiologists, the performance of the trained YOLOX model will be overestimated. So, the GTs generated by the human-in-the-loop approach cannot be applied in any test dataset. In fact, the GTs of all test datasets in our study were not generated by the human-in-the-loop approach and were instead manually annotated by two experts from the expert group together. The details are shown in **Table R1-1**.

We also performed pairwise comparisons to evaluate the performance of the junior radiologists, senior radiologists, and YOLOX model. **Table R1-5** shows the F1 score for the YOLOX model and six radiologists with an IoU-T of 0.5. We performed one-way analysis of variance (ANOVA) for the YOLOX model, junior radiologists, and senior radiologists and used the least significant difference (LSD) method for pairwise comparisons between groups (**Table R1-6 and R1-7**). The results show that the performance of the YOLOX model was superior to that of six radiologists and significantly better than that of the junior radiologists ($p=0.034$). We added the results to the revision manuscript.

(Continued on the next page)

Table R1-5 F1 score of the YOLOX model and six radiologists on the held-out test dataset with an IoU-T of 0.5

	YOLOX model	Junior 1	Junior 2	Junior 3	Senior 1	Senior 2	Senior 3
Atelectasis	0.286	0.457	0.600	0.545	0.609	0.590	0.525
Calcification	0.743	0.369	0.443	0.400	0.460	0.495	0.534
Consolidation	0.630	0.483	0.357	0.398	0.439	0.485	0.440
Effusion	0.695	0.197	0.396	0.281	0.273	0.334	0.198
Emphysema	0.646	0.698	0.775	0.597	0.800	0.744	0.701
Fibrosis	0.626	0.309	0.277	0.325	0.353	0.405	0.331
Fracture	0.786	0.411	0.436	0.573	0.453	0.580	0.581
Mass	0.712	0.634	0.646	0.699	0.645	0.814	0.782
Nodule	0.455	0.468	0.300	0.192	0.313	0.262	0.213
Pleural thickening	0.482	0.271	0.220	0.270	0.275	0.291	0.416
Pneumatoxis	0.637	0.311	0.376	0.490	0.465	0.539	0.394
Pneumothorax	0.803	0.667	0.689	0.746	0.627	0.688	0.752
Postoperative Metal	0.898	0.908	0.916	0.917	0.873	0.936	0.953
Venipuncture	0.911	0.919	0.926	0.958	0.924	0.920	0.915
Mean	0.665	0.507	0.525	0.528	0.536	0.577	0.552

Table R1-6 ANOVA for F1 score achieved by the YOLOX model, junior group and senior group on the held-out test dataset with an IoU-T of 0.5.

	Sum of Squares	df	Mean Square	F	p
Between Groups	0.220	2	0.110	2.319	0.104
Within Groups	4.507	95	0.047		
Total	4.727	97			

Table R1-7 Multiple comparisons among the YOLOX model, junior group, and senior group on the held-out test dataset with an IoU-T of 0.5.

(I) Group	(J) Group	Mean		p	95% Confidence Interval	
		Difference (I-J)	Std. Error		Lower Bound	Upper Bound
Junior group	Senior group	-0.035	0.048	0.461	-0.130	0.059
Senior group	YOLOX	-0.110	0.067	0.106	-0.243	0.024
YOLOX	Junior group	0.145*	0.067	0.034	0.011	0.278

*. The mean difference is significant at the 0.05 level.

- It will also be helpful to validate the accuracy of the bounding box annotations in the training set. Even though it may not affect the final performance since the model training may suppress the effects of some of the annotation errors.

Response: We completely agree with your opinion that the training set has a certain fault tolerance.

As we know, pathological results are often considered as the golden standard of abnormality. Thus, if our goal is to evaluate the accuracy of the annotations (bounding boxes) in the training dataset, we need to obtain the pathological results of the abnormalities on

CXRs. However, there are often no pathological results for abnormalities in CXRs, such as calcification, fibrosis, pleural thickening, etc, in routine clinical work. Therefore, it is difficult to evaluate the accuracy of the annotations in the training set.

In general, datasets are manually annotated by radiologists and can be considered the ground truth (GT) for deep learning [1]. However, it is time-consuming and difficult for radiologists to manually annotate the data due to the many CXRs in large datasets. To solve this problem, various auxiliary annotation methods have been proposed, such as extracting keywords directly from radiology reports through natural language processing (NLP). Unlike the identification and localization task in the proposed article, NLP is primarily used for medical image classification tasks. For example, references [1] and [2] used the NLP technique; however, the annotations extracted by NLP are not sufficiently accurate, as clearly noted in the above two articles, and the annotation accuracy also cannot be quantitatively evaluated in the above two articles. Some technical methods may be used to improve the annotation accuracy.

The human-in-the-loop approach is an effective annotation method that can be used for annotation in classification, segmentation, identification, and localization tasks. In this study, the training and validation sets were both annotated using the human-in-the-loop approach. The annotations obtained by the human-in-the-loop approach can only be considered pseudo annotations [3]. In this study, we used two radiologists with varying experience levels to manually correct the pseudo annotations. The checks by the two levels of radiologists are analogous to the routine workflow of radiologists interpreting CXRs. To ensure the standardization and consistency of the manual annotation results, we also formulated annotation rules and trained all radiologists involved in the annotations on these rules to ensure the annotation accuracy to the greatest extent. Therefore, the corrected annotations by the two radiologists can be considered the GT (ground truth) bounding boxes. The proposed dataset annotation method draws on the ideas of the following reference, which is also cited in the article. In reference [3], the authors developed a crowdsourced human-in-the-loop approach to segment cells. In this method, humans and algorithms work in tandem to produce accurate annotations, which means that the annotation results obtained by the human-in-the-loop approach can be considered the GTs in the following deep-learning tasks.

Additionally, reference [4] reported that “particularly because AI applications often involve large-scale and unstructured data, automated or technology-assisted human-in-the-loop methods are required to systematically address the data challenges.”

Thus, human-in-the-loop and other technical approaches can help address the challenge of annotating large-scale datasets. The annotation results of the training dataset using the human-in-the-loop approach can be considered the GT data for model training.

In addition, to accurately evaluate the performance of the deep model, the test set cannot be annotated with the human-in-the-loop approach, and this study strictly follows this rule. The experimental results on each test set show that the trained deep model performs better than the senior radiologists, which demonstrates the efficiency of the human-in-the-loop approach.

References

- [1] Wang C, et al. Development and validation of an abnormality-derived deep-learning diagnostic system for major respiratory diseases, *npj Digital Medicine*, 2022, 5: 124.

- [2] Guo Y, et al. Deep learning with weak annotation from diagnosis reports for detection of multiple head disorders: a prospective, multicentre study. *Lancet Digit Health*, 2022, doi: 10.1016/S2589-7500(22)00090-5.
- [3] Greenwald N F, et al. Whole-cell segmentation of tissue images with human-level performance using large-scale data annotation and deep learning. *Nature Biotechnology*, 2022, 40(4): 555-565.
- [4] Liang W X, et al. Advances, challenges and opportunities in creating data for trustworthy AI. *Nature Machine Intelligence* 8(2022):4.

- It will also be helpful to analyze the results of the external validation set in detail. For example, how the expert group will affect the performance. How are the external data varied from the training data (domain gaps), e.g., on each abnormality category?

Response: Thank you for this valuable suggestion.

We further analyzed the results of the external test dataset in the Discussion section of the manuscript as follows (see the yellow highlighted text in the revised manuscript):

The expert group manually annotated the GT bounding boxes of the external test dataset. The data in the external test dataset were not involved in the development of the model; therefore, these data do not affect the performance of the model.

The distributions of all abnormalities in each dataset are shown in **Table 1** and **Supplementary Table 4**. These two tables suggest that the proportion of nodules in external datasets A-D was higher than that in the CXR-AL14 dataset. In contrast, the proportions of pneumatosis, pneumothorax, postoperative metal and venipuncture were lower than those in the CXR-AL14 dataset, which may have resulted from the difference in the hospital level. Thus, the distribution of abnormalities in CXRs in four multicentre was different from those obtained at our hospital. However, the results in this study demonstrated that the distribution of abnormalities had little effect on the performance of the YOLOX model. In the multicentre validation results, the AP value and F1 score of pneumatosis and effusion were lower than those of the held-out test dataset. These results likely occurred because in the multicentre dataset, a small amount of pleural effusion was more common and the pneumatosis was small and scattered. In object detection, it is typically more difficult to identify small objects; thus, the performance of these two abnormalities achieved by the YOLOX model was reduced on multicentre test datasets.

- The intention of adopting YOLOX is questionable instead to other object detection frameworks since the series of YOLO detection methods are more outstanding in the real-time object detection task. However, its accuracy is primarily equivalent and often worse than the anchor-based/region-proposal-based methods, e.g., Faster-RCNN, etc. It is unclear about the design choices here.

Response: Thank you for your comments.

As we know, YOLO series networks belong to the one-stage method, while Faster-RCNN belongs to the two-stage method. The experimental results show that the two-stage framework consistently achieves good performance on the challenging COCO benchmark.

YOLO series models always pursue the optimal speed and accuracy trade-off in real-time applications. Based on the following references [1-3], with the development of target detection technology, the target detection performance of the YOLO series is also gradually improved. With recent advanced detection techniques (e.g., decoupled head, anchor-free, and advanced label assigning strategies), the YOLOX model produces substantially faster calculations and achieves considerably higher accuracies than other two-stage methods. Therefore, we chose the YOLOX architecture as the deep learning network to identify and localize multiple abnormalities in the CXRs. Furthermore, we trained the localization model for multiple abnormalities based on Faster-RCNN using the CXR-AL14 dataset and tested it on the held-out test dataset; the results are shown in **Figure R1-1**. We found that its performance was not superior to that of the proposed YOLOX model, as shown in **Figure R1-2**.

References

- [1] Ge Z, et al. YOLOX: exceeding YOLO series in 2021. rXiv preprint arXiv:2107.08430 (2021).
- [2] Lin T Y, Goyal P, Girshick R. Focal loss for dense object detection. IEEE Transactions on Pattern Analysis and Machine Intelligence, 2020, 42(2): 318-327.
- [3] Bochkovskiy A, Wang C Y, Liao H Y M. YOLOv4: Optimal speed and accuracy of object detection. arXiv: 2004.10934v1.

Figure R1-1 Performance of Faster-RCNN on the held-out test dataset with different IoU-Ts. (A) AP values of 14 abnormalities with an IoU-T of 0.5, (B) AP values of 14 abnormalities with an IoU-T of 0.3, and (C) AP values of 14 abnormalities with an IoU-T of 0.1.

(**Figure R1-2** can be found on the next page)

Figure R1-2 Performance comparison between the YOLOX model and the faster-RCNN model with different IoU-Ts.

Reviewer #2 (Remarks to the Author):

The authors make a keen observation that AI/ML-based diagnostic methods for chest x-rays tend to focus on a single pathology when more than one might exist. They also observe that sufficient training power is needed for every pathology to be able to accurately detect and localize the disease toward aiding the radiologist. They observe that heat maps are insufficiently accurate as a localization method.

To respond to the above, the authors built a large chest x-ray dataset with 14 pathologies. They trained a YOLOX localizer and detector for them. The performance of the detector is quite good, however, they have had to compromise on the IoU-T (threshold) values such that only 10% IoU is considered acceptable. Nonetheless, it is a move in the right direction.

Response: Thank you for your comment.

The setting of IoU-T has a strong impact on the evaluation of the YOLOX model performance. If the predicted bounding box overlaps with GT at this threshold, the bounding box predicted by the YOLOX model is considered correct. Therefore, we set three IoU-T values (0.1, 0.3, and 0.5) to evaluate the performance of the YOLOX model, and the model outperformed the senior radiologists at these three thresholds. Therefore, we did not consider the results of 10% IoU to be acceptable in the manuscript. But we maintain the opinion that 10% IoU can indicate that there is an abnormality in the bounding box area, which must be observed by radiologists. For the CAD of the CXRs, the multi-abnormality localization task is much more difficult than the classification task. The use of a deep-learning classification network can achieve only a CXR containing certain abnormalities (might achieve a better ROC) but cannot accurately localize the abnormality. Therefore, as you said, "it is a move in the right direction."

However, the manuscript leave much unsaid and in my opinion is not ready for publication.

Response: We have carefully revised the paper and added information according to the comments of all reviewers. We believe that the paper has been markedly improved as a result of these suggestions.

Criticisms/corrections/comments:

1. The structure of the manuscript is really weird. The authors are recommended to restructure content (and trim text) following a more standard structure
Introduction, Prior Art, Data, Methods, Experiments, Analysis, (Other contributions),
Conslusions, References

There is high redundancy in the material and significant cuts / edits to the text will still leave the impact in place.

Response: Thank you for your comment.

When we initially prepared the manuscript, we structured the content according to the structure you stated above. However, according to the *Nature Communications* instructions for authors, the paper must be structured like this: Title, Authors, Introduction, Results,

Discussion, Methods, Data Availability, Code Availability, References, etc. The instructions can be found on the following website: <https://www.nature.com/documents/ncomms-formatting-instructions.pdf>. Therefore, the paper was structured as required by *Nature Communications*.

We have made appropriate adjustments and reductions to the redundant content in the Methods section. However, some content was added in response to the comments of other reviewers.

2. Data set development - what was the inter-reader and intra-reader variability following the human-in-the-loop approach. What was the performance of the algorithm? How many iterations? How many images were reviewed per iteration? How many images per disease per reviewer? What is the overlap in the diseases per image, i.e., how many images have multiple diseases?

- what was the inter-reader and intra-reader variability following the human-in-the-loop approach.

Response: Thank you for your suggestions.

We observed the interreader and intrareader variability of the human-in-the-loop approach as follows.

First, 600 CXRs were randomly selected from the CXR-AL14 dataset (excluding the 8000 CXRs for training the preliminary model) and interpreted by the preliminary model. Then, each radiologist of the twelve radiologists was given 50 CXRs on average for correction (the process is completely consistent with the human-in-the-loop approach). Finally, we sent the 600 CXRs with corrected annotations to six experts for further correction. After each expert independently corrected all 600 CXRs for the first time, the IoU values of each abnormality between pairs of experts was calculated to observe the interreader variability. One month later, the annotations of all 600 CXRs corrected by the twelve radiologists were independently corrected by the six experts for a second time to evaluate the intrareader variability by calculating the IoU values of each abnormality between two annotations. Since the six experts played the final role in the creation of the CXR-AL14 dataset, we observed the interreader and intrareader variability only among these six experts.

It is well known that the larger the intersection over union (IoU) value is, the higher the overlap rate between two bounding boxes and the better their consistency [1] [2]. Referring to the method described in reference [1], we calculated the IoU value of all annotations to observe interreader and intrareader variability for each abnormality. **Table R2-1** shows that the average intrareader IoU value of each abnormality ranged from 0.773 to 0.992, and the average interreader IoU value of each abnormality ranged from 0.698 to 0.968.

In addition, we found that $\text{IoU}=0$ for a few abnormalities. This occurred in two situations. One was that certain abnormalities were annotated by one expert but not by another expert, and the other was that certain abnormalities were annotated differently by two experts. We

further counted the number of bounding boxes with IoU=0 for each abnormality. **Table R2-2** shows that there was no case with IoU=0 for most abnormalities. Only for several abnormalities, the maximal number was 14 in the interreader study (<0.023 per CXR) and the maximal number was 6 in the intrareader study (<0.01 per CXR). Therefore, the proposed results indicate good agreement of annotations for both interreader and intrareader.

The above content had been added to the revised manuscript.

Reference

[1] Maloca PM, et al. Validation of automated artificial intelligence segmentation of optical coherence tomography images. PLoS One. 2019 Aug 16; 14(8):e0220063.

[2] Rajaraman S, et al. Analyzing inter-reader variability affecting deep ensemble learning for COVID-19 detection in chest radiographs. PLoS One. 2020 Nov 12; 15(11):e0242301.

(Continued on the next page)

Table R2-1 IoU (mean ± SD) of all the bounding boxes for each abnormality on interreader or intrareader variability

	Atelectasis	Calcification	Consolidation	Effusion	Emphysema	Fibrosis	Fracture	Mass	Nodule	Pleural thickening	Pneumatosi s	Pneumothorax	Postoperative metal	Venipuncture	
Inter-reader (mean ±SD)	E 1 vs E 2	0.896±0.048	0.767±0.098	0.864±0.072	0.813±0.103	0.967±0.021	0.809±0.102	0.793±0.095	0.838±0.071	0.800±0.113	0.820±0.067	0.850±0.080	0.905±0.059	0.864±0.089	0.761±0.105
	E 1 vs E 3	0.826±0.030	0.730±0.103	0.801±0.094	0.771±0.106	0.944±0.022	0.755±0.094	0.736±0.088	0.859±0.085	0.733±0.107	0.799±0.081	0.820±0.101	0.880±0.057	0.835±0.078	0.748±0.096
	E 1 vs E 4	0.868±0.027	0.753±0.097	0.841±0.084	0.784±0.109	0.958±0.019	0.766±0.108	0.749±0.102	0.859±0.083	0.772±0.105	0.794±0.081	0.822±0.090	0.889±0.065	0.853±0.086	0.769±0.109
	E 1 vs E 5	0.849±0.065	0.742±0.113	0.822±0.084	0.801±0.099	0.950±0.023	0.789±0.099	0.760±0.092	0.882±0.037	0.773±0.113	0.792±0.087	0.812±0.105	0.880±0.061	0.855±0.078	0.743±0.101
	E 1 vs E 6	0.894±0.048	0.729±0.095	0.810±0.090	0.754±0.118	0.956±0.022	0.764±0.097	0.736±0.094	0.863±0.058	0.717±0.123	0.754±0.084	0.823±0.095	0.846±0.102	0.841±0.084	0.725±0.095
	E 2 vs E 3	0.868±0.017	0.699±0.104	0.802±0.101	0.772±0.109	0.944±0.025	0.744±0.116	0.791±0.071	0.872±0.069	0.748±0.108	0.807±0.074	0.831±0.094	0.900±0.071	0.812±0.080	0.779±0.083
	E 2 vs E 4	0.927±0.022	0.825±0.087	0.884±0.065	0.843±0.094	0.968±0.019	0.859±0.086	0.836±0.097	0.918±0.025	0.804±0.098	0.869±0.047	0.869±0.085	0.916±0.047	0.891±0.073	0.811±0.080
	E 2 vs E 5	0.911±0.044	0.803±0.105	0.875±0.089	0.842±0.104	0.956±0.023	0.867±0.085	0.842±0.091	0.829±0.084	0.805±0.094	0.852±0.103	0.871±0.080	0.915±0.061	0.895±0.073	0.787±0.082
	E 2 vs E 6	0.893±0.047	0.755±0.117	0.844±0.100	0.807±0.104	0.963±0.023	0.812±0.120	0.790±0.112	0.823±0.090	0.736±0.122	0.854±0.083	0.834±0.094	0.872±0.109	0.849±0.097	0.768±0.111
	E 3 vs E 4	0.867±0.054	0.701±0.113	0.784±0.098	0.729±0.118	0.947±0.023	0.705±0.113	0.750±0.093	0.883±0.057	0.762±0.098	0.775±0.074	0.788±0.124	0.884±0.068	0.814±0.075	0.748±0.073
	E 3 vs E 5	0.834±0.023	0.698±0.106	0.784±0.101	0.758±0.115	0.967±0.017	0.747±0.109	0.772±0.075	0.831±0.085	0.750±0.109	0.780±0.090	0.808±0.095	0.881±0.075	0.828±0.087	0.747±0.094
	E 3 vs E 6	0.825±0.034	0.714±0.107	0.785±0.101	0.723±0.129	0.936±0.026	0.733±0.107	0.737±0.088	0.857±0.084	0.755±0.110	0.766±0.083	0.804±0.109	0.853±0.114	0.855±0.085	0.739±0.076
	E 4 vs E 5	0.873±0.053	0.774±0.097	0.846±0.079	0.801±0.103	0.952±0.020	0.809±0.093	0.782±0.087	0.841±0.082	0.792±0.093	0.819±0.085	0.831±0.117	0.899±0.060	0.863±0.086	0.771±0.088
	E 4 vs E 6	0.860±0.034	0.749±0.119	0.848±0.091	0.805±0.106	0.964±0.021	0.809±0.116	0.751±0.113	0.843±0.067	0.739±0.120	0.820±0.084	0.831±0.101	0.872±0.112	0.852±0.089	0.776±0.124
	E 5 vs E 6	0.843±0.017	0.723±0.117	0.827±0.094	0.777±0.099	0.943±0.025	0.802±0.101	0.758±0.119	0.882±0.047	0.720±0.118	0.810±0.093	0.815±0.108	0.871±0.106	0.845±0.101	0.727±0.102
Intra-reader (mean ±SD)	E 1 vs E 1*	0.931±0.041	0.804±0.109	0.905±0.087	0.864±0.098	0.983±0.010	0.833±0.111	0.791±0.115	0.929±0.021	0.773±0.087	0.857±0.080	0.858±0.116	0.948±0.035	0.863±0.110	0.806±0.074
	E 2 vs E 2*	0.939±0.028	0.851±0.101	0.921±0.064	0.886±0.079	0.977±0.011	0.888±0.082	0.877±0.066	0.959±0.009	0.803±0.087	0.939±0.036	0.864±0.114	0.927±0.049	0.909±0.046	0.837±0.080
	E 3 vs E 3*	0.947±0.019	0.839±0.102	0.924±0.072	0.897±0.076	0.985±0.007	0.891±0.088	0.880±0.086	0.934±0.069	0.817±0.089	0.914±0.049	0.910±0.094	0.957±0.040	0.928±0.057	0.773±0.096
	E 4 vs E 4*	0.912±0.084	0.819±0.113	0.913±0.085	0.866±0.097	0.977±0.015	0.897±0.063	0.854±0.108	0.944±0.021	0.803±0.091	0.881±0.103	0.850±0.121	0.961±0.024	0.907±0.069	0.792±0.074
	E 5 vs E 5*	0.973±0.016	0.897±0.078	0.941±0.076	0.925±0.093	0.992±0.007	0.940±0.045	0.878±0.099	0.976±0.008	0.824±0.089	0.924±0.093	0.858±0.139	0.979±0.013	0.944±0.033	0.855±0.059
	E 6 vs E 6*	0.956±0.025	0.828±0.108	0.896±0.075	0.848±0.101	0.971±0.016	0.879±0.082	0.869±0.063	0.934±0.037	0.810±0.095	0.902±0.064	0.867±0.104	0.940±0.039	0.923±0.056	0.803±0.095

*, The twice annotations of each expert.

Table R2-2 Number of bounding boxes with IoU=0 for each abnormality on interreader or intrareader variability

	Atelectasis	Calcification	Consolidation	Effusion	Emphysema	Fibrosis	Fracture	Mass	Nodule	Pleural thickening	Pneumato- sis	Pneumoth- orax	Postoperat- ive metal	Venipunct- ure
Inter- reader (N=600)	E 1 vs E 2	0	1	2	0	0	2	0	0	3	0	0	0	0
	E 1 vs E 3	0	7	0	0	0	7	0	0	13	0	0	0	0
	E 1 vs E 4	0	2	2	0	0	2	0	0	10	0	0	0	0
	E 1 vs E 5	0	2	6	0	0	2	0	0	1	0	0	0	0
	E 1 vs E 6	0	4	3	0	0	2	0	0	11	0	0	0	0
	E 2 vs E 3	0	8	2	1	0	5	0	0	9	0	0	0	0
	E 2 vs E 4	0	0	0	0	0	0	0	0	7	0	0	0	0
	E 2 vs E 5	0	0	4	0	0	0	0	0	3	0	0	0	0
	E 2 vs E 6	0	2	1	0	0	0	0	0	9	0	0	0	0
	E 3 vs E 4	0	9	2	1	0	5	0	0	14	0	0	0	0
	E 3 vs E 5	0	9	6	1	0	5	0	0	12	0	0	0	0
	E 3 vs E 6	0	7	3	1	0	5	0	0	10	0	0	0	0
	E 4 vs E 5	0	0	4	0	0	0	0	0	10	0	0	0	0
	E 4 vs E 6	0	2	1	0	0	0	0	0	10	0	0	0	0
	E 5 vs E 6	0	2	5	0	0	0	0	0	10	0	0	0	0
Intra- reader (N=600)	E 1 vs E 1*	0	1	1	1	0	0	1	0	2	0	0	0	0
	E 2 vs E 2*	0	1	0	0	0	2	0	0	2	0	0	0	0
	E 3 vs E 3*	0	0	1	0	0	2	0	0	2	0	1	0	0
	E 4 vs E 4*	0	1	2	0	0	1	0	0	1	0	0	0	0
	E 5 vs E 5*	0	1	1	0	0	4	1	0	6	1	0	0	0
	E 6 vs E 6*	0	0	1	0	0	1	1	0	1	0	0	0	0

*. The twice annotations of each expert.

What was the performance of the algorithm?

Response: The performance of the updated deep model after each iteration was evaluated on the held-out test dataset, and the results are shown in **Figure R2-1**.

As the number of iterations increases, the number of annotated CXRs also increases, and the performance of the updated model gradually improves, as demonstrated in **Figure R2-1**. Note that the model with 0 iterations corresponds to the preliminary model. After seven iterations, the creation of CXR-AL14 dataset was completed, and the updated model after the seventh iteration was just the YOLOX model in the proposed framework, which achieved excellent performance. The results also demonstrated that the human-in-the-loop approach is effective for the annotation of the training dataset. The above content has been added to the revised manuscript.

Figure R2-1 Performance of the updated model with different IoU-Ts after each iteration.

How many iterations?

Response: After seven iterations, the CXR-AL14 dataset was created, as described in **Part 4.4**, page18, line21.

How many images were reviewed per iteration?

Response: We counted the number of CXRs per iteration during the creation of the CXR-AL14 dataset, which is shown in **Table R2-3** below. We add this table to the revised appendix (*Supplementary Table 25*).

Table R2-3 The number of CXRs per iteration during the construction of the CXR-AL14 dataset

	Preliminary model	Iteration 1	Iteration 2	Iteration 3	Iteration 4	Iteration 5	Iteration 6	Iteration 7	Total
CXRs	8000	24684	23761	24118	22186	23574	21694	17971	165988

(Continued on the next page)

How many images per disease per reviewer?

Response: We determined the number of abnormalities annotated by each expert in the process of constructing the CXR-AL14 dataset, as shown in **Table R2-4**. Each expert annotated more than 37,000 bounding boxes. In addition, the expert group discussed and annotated 8000 CXRs together, and these CXRs were used to train the preliminary model. We added this table to the revised appendix (*Supplementary Table 26*).

Table R2-4 Number of CXRs and each abnormality annotated by each expert.

	Expert group	Expert 1	Expert 2	Expert 3	Expert 4	Expert 5	Expert 6	Total
CXRs	8000	26329	26329	26332	26332	26332	26334	165988
Atelectasis	17	61	61	57	52	36	74	358
Calcification	1640	6278	5471	5254	5441	5015	4255	33354
Consolidation	1002	3533	3281	3135	2921	2797	4597	21266
Effusion	2015	6847	6488	6854	6421	6519	8967	44111
Emphysema	1159	3779	3966	4057	4027	3718	2884	23590
Fibrosis	750	2835	2486	2452	2490	2301	3373	16687
Fracture	575	2328	1927	1907	1793	1564	2538	12632
Mass	132	469	436	416	394	441	710	2998
Nodule	2170	8724	6518	6620	7771	7131	7043	45977
Pleural thickening	336	1685	1084	1218	962	872	1236	7393
Pneumatoxis	259	461	739	839	1068	1125	2462	6953
Pneumothorax	313	715	941	910	1002	1094	2089	7064
Postoperative Metal	866	3679	2629	2644	2884	2970	4257	19929
Venipuncture	482	894	1230	1472	1784	2004	3666	11532
Total	11716	42288	37257	37835	39010	37587	48151	253844

What is the overlap in the diseases per image, i.e., how many images have multiple diseases?

Response: We counted the number of CXRs containing more than one abnormality in the CXR-AL14 dataset. There were 47,016 CXRs with one abnormality, 28,683 with two abnormalities, and 27,205 with multiple abnormalities in the CXR-AL14 dataset (see **Table R2-5** below). We added the text to the manuscript and this table to the revised appendix (*Supplementary Table 24*).

Table R2-5 The number of CXRs with multiple abnormalities in the CXR-AL14 dataset

	Normal	single abnormality	2 abnormalities	3 abnormalities	4 abnormalities	5 abnormalities	6 abnormalities	7 abnormalities	8 abnormalities	Total
CXRs	63084	47016	28683	16369	7546	2537	613	118	22	165988

3. What is the performance of the algorithm as a function of the variability inherent in the dataset?

Response: Thank you for your meaningful comment.

Notably, the loss function values effectively demonstrate the performance of the model. In addition, the epoch value was an important hyperparameter for the training of the YOLOX

network. An epoch value that is too small could lead to underfitting of the model, while an epoch value that is too large could lead to overfitting of the model. The relationship between the loss values and the epoch values during the training of the YOLOX model is given in **Figure R2-2**. As the number of epochs increased, the values of the loss function on both the training and validation sets gradually decreased. When the number of epochs exceeded 27, the loss function values on the two datasets tended to be stable, which indicated that the network training had converged. The contents and the table have been added to the revised manuscript.

Figure R2-2 Loss curves of the YOLOX model with different epochs.

4. Why is the algorithm performing poorly for two diseases?

Response: This is a very good question.

In this study, the performance of the YOLOX model was poor in atelectasis and pneumatosis on the multicentre test dataset. There are two main reasons why the YOLOX model does not perform well with these two abnormalities. First and foremost, the sample size of atelectasis in the training dataset was insufficient (only 358 atelectasis bounding boxes in the CXR-AL14 dataset), which limited to train a model with good performance in localizing atelectasis. Therefore, the AP values of atelectasis achieved by the YOLOX model were only 0.152 and 0.083 with an IoU-T of 0.5 on the held-out and multicentre A test datasets, respectively.

What's more, in general, abnormalities with small size, blurring boundaries, and multiple scattered abnormalities were difficult to be accurately identified and localized. In multicentre test datasets, small and scattered pneumatosis was more common. Thus, the performance of the YOLOX model in localizing small and scattered pneumatosis was also limited.

We added the above contents to the Discussion section (see the yellow highlighted text in the revised manuscript).

(Continued on the next page)

5. Is it important to describe the tool that has been developed? If the tool must be a part of the manuscript, what is its performance in active use? What approximations / engineering was done to ensure quality control for the tool?

Response: We greatly appreciate your excellent questions.

The ultimate goal of our research is to develop a one-stop diagnostic tool for CXRs, which is completely consistent with the workflow of radiologists. Visualization of deep learning models is a critical step for clinical application. So, X-ray Eye software was developed according to the radiology department workflow in the study, in order to preliminary validate the clinical feasibility of our framework. Furthermore, prospective test demonstrated the proposed framework achieved good performance and clinical application prospects. However, considerable work must be performed before this software can be used in clinical practice. As you said, more quality control (e.g., approximation/engineering) is still required for the tool. In future research, we intend to enlarge the multicentre sample size and use these data during model training, further optimize the model, and conduct more extensive quality control and tests for actual clinical work. Therefore, in this paper, the X-ray eye software development was described briefly to demonstrate the feasibility and potential clinical applicability of the proposed framework.

Reviewer #3 (Remarks to the Author):

The aim of this paper is two folds for identifying and localizing 14 abnormalities and calculating the cardiothoracic ratio simultaneously with about 165 thousand CXRs and 253 thousand bounding boxes. This is quite interesting and clinical relevant topics. However, there are several concerns on a lack of technical novelty and the reproducibility of this study. I recommend this paper would be better for radiology related clinical journals.

Response: Thank you for these positive comments and encouragement.

Just as your comment, automatic identification and localization for multiple abnormalities in CXRs is a quite interesting and clinically relevant topic. Automatically identification and localization for multiple abnormalities in CXRs is completely consistent with the review process of radiologists. Due to the lack of large-scale CXR datasets containing bounding boxes, most previous studies tried to attain the aim through classification networks. However, model trained by classification networks cannot provide accurate localization of the abnormalities in CXRs and is not truly suitable for routine clinical setting. To the best of our knowledge, few studies have applied deep learning localization networks to accurately identify and localize multiple abnormalities because large-scale CXR datasets containing bounding boxes are unavailable. Thus, a large-scale CXR dataset with GT bounding boxes (CXR-AL14) was created in this study. Based on the created CXR dataset, the YOLOX model with high performance was developed to identify and localize multiple abnormalities in CXRs, which was completely consistent with the interpretation process of radiologists. The development of the X-ray Eye software on the basis of the framework also preliminarily showed good clinical application prospect of our framework. To the best of our knowledge, the CXR-AL14 dataset is the largest CXR dataset with GT bounding boxes in the world, which would significantly promote the research of the localization task. The framework in the study is not only more suitable for routine clinical work than AI model reported previously, but also own good performance, generalization ability and strong clinical application prospect.

Regarding the reproducibility of this framework, the results of the 5-fold cross-validation, multicentre tests and prospective test demonstrate that the framework has good repeatability, applicability and robustness.

As you said, we have considered publication in a radiology journal. However, after discussion by all the authors, we decided to publish our work in a more comprehensive journal. First, this study involved interdisciplinary research. Accurate identification and localization of multiple abnormalities and the calculation of the CTR with one framework can truly meet the daily clinical needs of radiologists. Second, we constructed a large dataset of CXRs with bounding boxes. We believe that the creation of this dataset may promote deep learning-based research on the localization of multiple abnormalities. Thus, this study is of interest to deep learning researchers, clinical radiologists, chest-related clinicians, and emergency department doctors. The benefit of this article would be limited if the work is published in a radiology journal. *Nature Communications* is a highly influential and comprehensive multidisciplinary journal, and its papers have been openly reviewed.

Therefore, we think that our manuscript is particularly suitable for *Nature Communications*, an international journal, and will be of great interest to its readers.

1. Is this dataset (CXR-AL14) collected in consecutive manner? How do you differentiate AP CXR vs PA CXR? In addition, there is a lack of details (selection and exclusion criteria, duration, consecutive or not, etc) on dataset of external and prospective validations.

Response: Thank you for your helpful question.

-Is this dataset (CXR-AL14) collected in consecutive manner?

Response: Yes, the CXRs in the CXR-AL14 dataset were collected continuously between August 2011 and December 2021 and filtered with exclusion criteria. We added this information to the revised manuscript.

-How do you differentiate AP CXR vs PA CXR?

Response: By reading the DICOM information of the CXRs, the AP and PA can be easily distinguished.

-In addition, there is a lack of details (selection and exclusion criteria, duration, consecutive or not, etc) on dataset of external and prospective validations.

Response: The CXRs in the external and prospective test datasets were collected randomly, not consecutively, from the PACS of their respective hospitals, and the exclusion criteria were consistent with the CXR-AL14 dataset. We added this information to the revised manuscript. Details for the collection times are shown in *Supplementary Table 4*.

2. There is no external or cross-validation for evaluating the reproducibility of this method. Especially, leave-one-out methods could predict over-estimation. Please use 3-5 fold cross-validation or external validation

Response: Thank you for your valuable comments.

A 5-fold cross-validation was performed on the CXR-AL14 dataset, in which five models were generated. The performance of the five models in the 5-fold cross-validation with an IoU-T of 0.5 is shown in **Figure R3-1**. Additionally, the five models were evaluated on the held-out test dataset, and the corresponding results with an IoU-T of 0.5 are illustrated in **Figure R3-2**. Based on the experimental results, the models trained on the CXR-AL14 dataset achieved similar performance, demonstrating the good stability and repeatability of the proposed framework. We added these results to the revised manuscript.

(Continued on the next page)

Figure R3-1 Performance of the five trained models generated by 5-fold cross-validation with an IoU-T of 0.5

Figure R3-2 Performance of the five trained models generated by 5-fold cross-validation on the held-out test dataset with an IoU-T of 0.5

3. In case of anotation rule 2 (if the edge of one abnormality is obscured or ...) or especially in case of disageement among radiologists, how to confirm according to the knowledge and experience of radiologists. Describe in detail.

Response: Thank you for this valuable comment. We provide an example below to illustrate annotation rule 2, (see **Figure R3-3** on the next page).

Figure R3-3 An example to illustrate annotation rule 2.

In this CXR, a moderate left pleural effusion sheltered the left hemidiaphragm and costophrenic angle. Thus, the lower edge of the bounding box of the effusion cannot easily be determined. Radiologists should use their knowledge and experience to determine the position of the left hemidiaphragm and costophrenic angle by referring to their position on the right side (shown as the dotted line in the figure), and further annotate the lower boundary of the left pleural effusion. We added this to the revised manuscript.

4. In case of human in the loop approach, please describe the accuracy improvements in each iteration.

Response: Thank you for your helpful comment.

The performance of the updated deep model after each iteration was evaluated on the held-out test dataset. The results are shown in **Figure R3-4**.

As the number of iterations increases, the number of annotated CXRs also increases, and the performance of the updated model gradually improves, as shown in **Figure R3-4**. Note that the model with 0 iteration corresponds to the preliminary model. After seven iterations, the creation of the CXR-AL14 dataset was completed, and the updated model after the seventh iteration was just the YOLOX model in the proposed framework, which achieved excellent performance. **Figure R3-4** also shows that the human-in-the-loop approach was effective when annotating the training dataset. We added this in the revised manuscript.

(Continued on the next page)

Figure R3-4 Performance of the updated model with different IoU-Ts after each iteration.

5. There is a lack of methodological originality.

Response: Just as your first item of comment, automatic identification and localization for multiple abnormalities in CXRs is a quite interesting and clinically relevant topic. Automatically identification and localization for multiple abnormalities in CXRs is completely consistent with the review process of radiologists. Due to the lack of large-scale CXR datasets containing bounding boxes, most previous studies tried to attain the aim through classification networks. However, model trained by classification networks cannot provide accurate localization of the abnormalities in CXRs and is not truly suitable for routine clinical setting. To the best of our knowledge, few studies have applied deep learning localization networks to accurately identify and localize multiple abnormalities because large-scale CXR datasets containing bounding boxes are unavailable. Thus, a large-scale CXR dataset with GT bounding boxes (CXR-AL14) was created in this study. Based on the created CXR dataset, the YOLOX model with high performance was developed to identify and localize multiple abnormalities in CXRs, which was completely consistent with the interpretation process of radiologists. The development of the X-ray Eye software on the basis of the framework also preliminarily showed good clinical application prospect of our framework. To the best of our knowledge, the CXR-AL14 dataset is the largest CXR dataset with GT bounding boxes in the world, which would significantly promote the research of the localization task. The framework in the study is not only more suitable for routine clinical work than AI model reported previously, but also own good performance, generalization ability and strong clinical application prospect.

In addition, we also compared the performance achieved by the YOLOX model and the Faster-RCNN model (see **Figure R3-5** next page). It is obvious that the performance of the YOLOX model is significantly better than the Faster-RCNN model.

In the future, we will continue to increase the amount of data in the CXR-AL14 dataset and improve the localization network to enhance the localization performance. The ultimate purpose of our research is to apply the framework in routine clinical practice.

(Continued on the next page)

Figure R3-5 Performance comparison between the YOLOX model and the Faster-RCNN model with different IoU-Ts.

6. There is no stress test, ablation and parameter changing studies for evaluating optimization of models.

Response: Thank you for your comments.

We performed a quality control stress test for the YOLOX model on the held-out test dataset. The quality of CXRs is primarily determined by the brightness and contrast of the CXRs. Thus, we tested the performance of the YOLOX model with varying degrees of brightness and contrast changes on the held-out test dataset. Note that negative and positive brightness and contrast changes were tested, with increments and decrements of 0.1, where 1.0 was the baseline (**Figure R3-6**). It was obvious that small changes in brightness and contrast had little effect on the performance of the YOLOX model; however, large changes in brightness and contrast had marked impacts on the model. However, in routine clinical practice, there are few CXRs with particularly poor brightness and contrast. In addition, the brightness and contrast of CXRs in each test dataset may differ because the CXRs were generated by various X-ray equipment, and the YOLOX model achieved good performance on each test dataset. These results demonstrate that the model could effectively handle changes in brightness and contrast, showing good generalizability. We added these contents to the revised manuscript.

(Continued on the next page)

Figure R3-6 Performance curves of the YOLOX model under different changes of brightness or contrast.

In this study, the YOLOX architecture was directly used to train the deep model to identify and localize abnormalities, without any additional structures added to the YOLOX architecture; thus, there is no issue in the ablation experiment. Improvements in the network architecture are important issues that must be addressed in future research.

It should be noted that the adjustment of super parameters was completed in the process of model training and tuning to optimize the trained model. The values of the loss function effectively demonstrate the performance of the model, and the epoch value was an important hyperparameter for the training of the YOLOX network. An epoch value that is too small could lead to underfitting of the model, while an epoch value that is too large could lead to overfitting of the model. The relationship between the loss values and the epoch values during the training of the YOLOX model is given in **Figure R3-7**. As the number of epochs increased, the values of the loss function on both the training and validation sets gradually decreased. When the number of epochs exceeded 27, the loss function values on the two datasets tended to be stable, which indicated that the network training had converged. We added these contents to the revised manuscript.

Figure R3-7 Loss curves of the YOLOX model with different numbers of epochs.

Reviewer #4 (Remarks to the Author):

This study developed and validated a YOLOX model for identifying and localizing 14 abnormalities and assessing cardiomegaly in Chest X-ray with a large-scale datasets containing 165,988 radiographs and 253844 bounding boxes.

1. Bounding box segmentation has some difficulty for labeling pulmonary lesions on chest radiograph because a box cannot follow anatomical structures or lesion shape. Pixel-based annotation is usually adopted for commercially available AI-based diagnosis-software for chest radiographs. For example, in other study (reference #2), they developed a high-performance model with 146717 chest radiographs with pixel-based annotation for 10 abnormalities.

Response: Highly appreciate this excellent comment, we quite agree with you.

As you said, the bounding box segmentation has some difficulty labelling pulmonary lesions in CXRs. However, bounding boxes are suitable for deep learning localization tasks. In clinical work, radiologists' interpretation of CXR can typically be divided into four processes. First and foremost, clinically relevant abnormalities should be identified and localized. This perspective was also mentioned by reference #2. In our study, the core problem we investigated is the identification and localization of multiple abnormalities. To achieve this task, it is necessary to train the model on a large CXR dataset containing various abnormality bounding boxes rather than segmentation of the lung lesions. The most common application of deep learning segmentation models is the quantitative calculation of abnormalities, such as the quantification of cerebral haemorrhage and the segmentation or extraction of tumour tissue. Heart-lung segmentation was also applied in the calculation of CTR in our study.

In reference #2, although the researchers performed voxel-based segmentation of the lesions, they still used a deep learning classification network (ResNet34-based neural network) to train model, as shown in **Figure R4-1** (cited from the Supplementary Materials of reference #2). Segmented lesions can improve the accuracy of classification tasks, particularly the accuracy of generated heatmaps. However, the primary purpose of this research was still to realize the localization of multiple abnormalities through classification networks.

In fact, the real number of annotations for 10 abnormalities was 41,698, not 146,717 in reference #2. However, in our study, the number of bounding boxes for 14 abnormalities was 253,844, and a localization network was used (different from classification networks in reference #2). Such a large scale of bounding boxes is sufficient to develop a deep learning localization model (YOLOX) with good performance.

(Continued on the next page)

Figure R4-1 Architecture of the DLAD-10 algorithm (cited from Supplementary Figures of reference #2).

2. IOU-threshold is a usual method to evaluate the accuracy of YOLOX model. However, qualitative evaluation of lesion detection on chest radiograph by expert radiologists might be more accurate method to evaluate the performance of the model. It could be done on external validation process.

Response: We completely agree with your comments.

In this study, both the classification and localization results are simultaneously provided by the YOLOX model. In order to accurately evaluate the performance of the YOLOX model, the annotations of all test datasets, including the held-out test dataset, multicentre test dataset, recombination test dataset (700 CXRs were randomly selected from each multicentre, a total of 2,800 CXRs in this dataset) and prospective test dataset, were performed manually by experts, which are considered the ground truth (bounding boxes).

As we know, IoU is an effective indicator to measure the degree of overlap between the predicted bounding box and the GT one. To evaluate the YOLOX model, different from the performance evaluation in the medical image segmentation task, the IoU is not a final index for performance evaluation but a premise of evaluation. In this study, the IoU threshold (IoU-T) is a preset value. For each predicted box, an IoU value can be calculated in combination with its GT bounding box, and only those predicted boxes with IoU values higher than the preset threshold (IoU-T) will be included in the calculation of subsequent indicators, such as precision, recall, and AP. Therefore, the IoU-T value directly affects the final evaluation results of YOLOX and the six radiologists. Obviously, the higher the IoU-T value is, the poorer the final performance evaluation result. Similarly, the lower the IoU-T value is, the better the final performance evaluation result. In this study, IoU-T is set to 0.1, 0.3 and 0.5, where IoU-T=0.5 is a harsh threshold in the field of object detection, and IoU-T=0.1 is a relatively low threshold. When IoU-T=0.1, the performance evaluation primarily focuses on the accuracy of the predicted label of each abnormality (i.e., qualitative evaluation of lesion detection on CXRs). As reported in this article, when IoU-T=0.1, the performance of the YOLOX model is still better than that of the best senior radiologists; however, its advantages are weak, and the performance difference is small, where the F1 value of the YOLOX model is 0.747 and the F1 value of the best senior radiologist is 0.737. Thus, the result of IoU-T=0.1 in this

article can describe the qualitative evaluation of less detection on CXRs. In this situation, the performance of the YOLOX model is comparable to that of senior radiologists.

There have been many qualitative evaluation studies on classifying abnormalities in CXRs. As mentioned in reference [1], in clinical practice, visual evidence that supports the classification result, such as spatial localization of sites of abnormalities, is indispensable in clinical diagnoses and provides interpretability and insights. Therefore, it is critical that the image analysis method provides both classification results and the associated visual evidence with high accuracy. Therefore, even if it is a qualitative evaluation, it is more reliable to use the results of IoU=0.1 in this study for qualitative evaluations than to perform qualitative evaluations without considering any localization, as done in previous work.

Reference:

[1] Zhe, L., et al. Thoracic Disease Identification and Localization with Limited Supervision. in 2018 IEEE/CVF Conference on Computer Vision and Pattern Recognition 8290-8299 (2018)

3. False positive rate (the number of false positive findings per image) is an important index to see how much attention and effort would be additionally needed to differentiate false positive lesions to read chest radiographs in workflow if the software is adopted in clinical reading session.

Response: Thank you for your comments.

We calculated the number of false positive (FP) findings for each abnormality in each CXR in all test datasets with different IoU-T values and determined the average number of false positive findings of all abnormalities in each CXR, which were shown in **Table R4-1**. The table was also added to the Supplementary Material (*Supplementary Table 20*). This table suggests that nodules accounted for the largest proportion of false-positive findings among all abnormalities (e.g., 53.23% in external test dataset B with an IoU-T of 0.3). However, the average number of FP findings of all abnormalities in each CXR was less than one with an IoU-T of 0.5. Thus, we believe that the YOLOX model has good performance in controlling false-positive findings and will not increase the workload of radiologists. The content was added to the revised manuscript.

(Continued on the next page)

Table R4-1 The number of false positive findings of each abnormality on different test datasets with different IoU-T values

Test dataset	IoU-T	Atelectasis	Calcification	Consolidation	Effusion	Emphysema	Fibrosis	Fracture	Mass	Nodule	Pleural thickening	Pneumatois	Pneumothorax	Postoperative metal	Venipuncture	Total	Average FP per CXR
Held-out (n=6000)	0.1	13	146	241	275	329	203	99	30	682	79	48	13	14	17	2189	0.365
	0.3	13	195	261	266	329	208	99	32	816	117	43	17	16	16	2428	0.405
	0.5	6	301	254	426	329	244	117	27	1245	67	72	23	28	27	3166	0.528
External dataset A (n=2978)	0.1	3	50	62	92	58	45	20	5	284	29	6	0	13	1	668	0.224
	0.3	3	62	67	96	58	52	20	5	308	30	9	1	13	1	725	0.243
	0.5	3	90	81	146	58	62	38	3	423	19	8	5	13	1	950	0.319
External dataset B (n=2633)	0.1	6	53	112	70	67	79	23	11	429	28	7	4	2	0	891	0.338
	0.3	1	60	124	79	67	39	16	11	503	29	8	6	2	0	945	0.359
	0.5	1	89	160	118	67	52	17	11	646	45	5	7	2	0	1220	0.463
External dataset C (n=2651)	0.1	3	142	285	138	324	107	63	19	870	46	6	7	1	3	2014	0.760
	0.3	4	159	307	145	324	95	65	24	919	65	6	7	1	3	2124	0.801
	0.5	4	208	354	209	324	113	69	26	1142	56	6	8	1	0	2520	0.951
External dataset D (n=2683)	0.1	4	118	120	96	299	35	46	19	742	24	13	6	0	1	1523	0.568
	0.3	4	91	128	100	299	38	47	19	783	26	16	6	0	1	1558	0.581
	0.5	5	119	141	142	299	52	54	19	969	33	24	8	0	1	1866	0.695
Recombination test dataset (n=2800)	0.1	11	73	168	113	134	97	26	13	597	60	2	4	5	1	1304	0.466
	0.3	12	87	193	124	134	109	26	14	641	61	3	4	5	1	1414	0.505
	0.5	9	111	217	185	134	118	28	17	835	77	3	5	5	1	1745	0.623
Prospective dataset (n=1022)	0.1	0	42	34	78	78	27	19	2	100	17	9	2	7	6	421	0.412
	0.3	0	55	35	82	78	27	19	2	105	17	13	3	7	7	450	0.440
	0.5	0	89	34	114	78	37	22	2	130	19	16	8	8	9	566	0.554

4. Comparison of performance of the YOLOX model with the radiologists better be conducted in external test sets as well. With the validation dataset from the one hospital which is the same source of training dataset may have some performance gain to YOLOX model than radiologists.

Response: Thank you for this meaningful suggestion.

We compared the performance of the YOLOX model with that of three junior and three senior radiologists on the external test dataset (recombination test dataset) as follows. First, 700 CXRs from each multicentre were randomly selected to construct the recombination test dataset (a total of 2800 CXRs). General information and the distribution of all abnormalities in the recombination test dataset was shown in *Supplementary Table 4*. Similar with the held-out test dataset, the same three senior radiologists and three junior radiologists independently annotated the above all 2800 CXRs according to formulated annotation principles. Then, we compared the performance among junior radiologists, and senior radiologists and the YOLOX model.

The YOLOX model also showed good performance, which was also superior to the senior radiologists. With an IoU-T of 0.5, the mean precision, recall and F1-score obtained by the YOLOX model were 0.640, 0.639 and 0.632, respectively, which were higher than those of the best senior radiologist, whose values were 0.632, 0.555 and 0.586, respectively (**Table R4-2/R4-3**). **Figure R4-2** shows that the performance of the YOLOX model was better than that of radiologists for most abnormalities, except for atelectasis and emphysema. The performance on the recombination test dataset was similar to the results obtained on the held-out test dataset.

We added the above content to the revised manuscript.

(Continued on the next page)

Figure R4-2 Performance comparison between the YOLOX model and radiologists on the recombination test dataset with an IoU-T of 0.5.

(Continued on the next page)

Table R4-2 Performance comparison among the YOLOX model, the best junior radiologist and the best senior radiologist on the recombination test dataset with an IoU-T of 0.5.

	YOLOX			Junior 3 (Best)			Senior 2 (Best)						
	Precision	Recall	F1	Precision	P-value	Recall	P-value	F1	Precision	P-value	Recall	P-value	F1
Atelectasis	0.471 (0.262-0.690)	0.615 (0.355-0.823)	0.533	0.667 (0.354-0.879)	0.340	0.462 (0.232-0.709)	1.000	0.545	0.700 (0.397-0.892)	0.247	0.538 (0.291-0.768)	1.000	0.609
Calcification	0.801 (0.766-0.832)	0.746 (0.709-0.779)	0.772	0.668 (0.624-0.709)	0.000	0.518 (0.478-0.558)	0.000	0.584	0.672 (0.633-0.709)	0.000	0.659 (0.620-0.696)	0.000	0.666
Consolidation	0.569 (0.526-0.612)	0.671 (0.625-0.713)	0.616	0.544 (0.499-0.588)	0.417	0.610 (0.563-0.655)	0.064	0.575	0.624 (0.573-0.673)	0.108	0.516 (0.469-0.563)	0.001	0.565
Effusion	0.389 (0.336-0.445)	0.490 (0.427-0.552)	0.434	0.319 (0.262-0.381)	0.092	0.307 (0.252-0.368)	0.000	0.313	0.419 (0.353-0.487)	0.510	0.353 (0.295-0.415)	0.002	0.383
Emphysema	0.660 (0.612-0.705)	0.720 (0.672-0.764)	0.689	0.838 (0.795-0.874)	0.000	0.776 (0.730-0.816)	0.105	0.806	0.828 (0.782-0.866)	0.000	0.709 (0.660-0.754)	0.799	0.764
Fibrosis	0.590 (0.533-0.646)	0.601 (0.543-0.656)	0.595	0.469 (0.410-0.528)	0.004	0.449 (0.392-0.507)	0.000	0.458	0.705 (0.625-0.775)	0.022	0.346 (0.293-0.403)	0.000	0.464
Fracture	0.861 (0.806-0.902)	0.762 (0.703-0.813)	0.808	0.868 (0.794-0.919)	0.848	0.436 (0.373-0.501)	0.000	0.581	0.753 (0.687-0.809)	0.007	0.630 (0.565-0.690)	0.000	0.686
Mass	0.605 (0.456-0.736)	0.722 (0.560-0.842)	0.658	0.595 (0.435-0.737)	0.927	0.611 (0.449-0.752)	0.227	0.603	0.719 (0.546-0.844)	0.304	0.639 (0.476-0.775)	0.508	0.676
Nodule	0.417 (0.392-0.443)	0.468 (0.441-0.495)	0.441	0.411 (0.380-0.443)	0.789	0.306 (0.281-0.331)	0.000	0.351	0.291 (0.269-0.314)	0.000	0.368 (0.342-0.395)	0.001	0.325
Pleural thickening	0.506 (0.429-0.584)	0.612 (0.526-0.692)	0.554	0.379 (0.291-0.475)	0.043	0.302 (0.230-0.386)	0.000	0.336	0.430 (0.340-0.525)	0.222	0.357 (0.279-0.442)	0.000	0.390
Pneumatois	0.500 (0.188-0.812)	0.231 (0.082-0.503)	0.316	0.273 (0.097-0.566)	0.349	0.231 (0.082-0.503)	1.000	0.250	0.300 (0.108-0.603)	0.424	0.231 (0.082-0.503)	1.000	0.261
Pneumothorax	0.783 (0.581-0.903)	0.581 (0.408-0.736)	0.667	0.652 (0.449-0.812)	0.326	0.484 (0.320-0.652)	0.774	0.556	0.618 (0.450-0.761)	0.189	0.677 (0.501-0.814)	0.549	0.646
Postoperative Metal	0.891 (0.770-0.953)	0.932 (0.818-0.977)	0.911	0.896 (0.778-0.955)	0.943	0.977 (0.882-0.996)	0.625	0.935	0.929 (0.810-0.975)	0.544	0.886 (0.760-0.950)	1.000	0.907
Venipuncture	0.923 (0.667-0.986)	0.800 (0.548-0.930)	0.857	0.933 (0.702-0.988)	0.916	0.933 (0.702-0.988)	0.500	0.933	0.867 (0.621-0.963)	0.630	0.867 (0.621-0.963)	1.000	0.867
Mean	0.640	0.639	0.632	0.608	-	0.529	-	0.559	0.632	-	0.555	-	0.586

Table R4-3 Performance of the other four radiologists on the recombination test dataset with an IoU-T of 0.5.

	Junior 1			Junior 2			Senior 1			Senior 3		
	Precision	Recall	F1	Precision	Recall	F1	Precision	Recall	F1	Precision	Recall	F1
Atelectasis	0.615 (0.355-0.823)	0.615 (0.355-0.823)	0.615	0.636 (0.354-0.848)	0.538 (0.291-0.768)	0.583	0.571 (0.326-0.786)	0.615 (0.355-0.823)	0.593	0.636 (0.354-0.848)	0.538 (0.291-0.768)	0.583
Calcification	0.723 (0.675-0.766)	0.445 (0.405-0.485)	0.551	0.725 (0.683-0.764)	0.557 (0.517-0.596)	0.630	0.695 (0.652-0.734)	0.570 (0.530-0.609)	0.626	0.671 (0.629-0.711)	0.564 (0.524-0.603)	0.613
Consolidation	0.548 (0.503-0.592)	0.603 (0.556-0.648)	0.574	0.559 (0.515-0.603)	0.629 (0.582-0.673)	0.592	0.566 (0.519-0.611)	0.584 (0.537-0.630)	0.575	0.709 (0.658-0.755)	0.551 (0.504-0.598)	0.620
Effusion	0.495 (0.424-0.566)	0.386 (0.327-0.449)	0.434	0.437 (0.362-0.515)	0.286 (0.233-0.346)	0.346	0.484 (0.413-0.555)	0.369 (0.311-0.432)	0.419	0.455 (0.375-0.536)	0.270 (0.218-0.329)	0.339
Emphysema	0.878 (0.836-0.911)	0.701 (0.652-0.746)	0.780	0.871 (0.827-0.906)	0.676 (0.626-0.722)	0.761	0.881 (0.841-0.912)	0.781 (0.736-0.821)	0.828	0.886 (0.844-0.918)	0.709 (0.660-0.754)	0.788
Fibrosis	0.720 (0.638-0.789)	0.336 (0.283-0.393)	0.458	0.493 (0.429-0.558)	0.396 (0.341-0.454)	0.439	0.517 (0.449-0.584)	0.382 (0.327-0.439)	0.439	0.547 (0.486-0.607)	0.491 (0.433-0.549)	0.518
Fracture	0.782 (0.702-0.846)	0.427 (0.365-0.492)	0.553	0.821 (0.741-0.880)	0.423 (0.360-0.488)	0.558	0.808 (0.742-0.861)	0.595 (0.530-0.656)	0.685	0.656 (0.589-0.716)	0.612 (0.548-0.673)	0.633
Mass	0.739 (0.535-0.875)	0.472 (0.320-0.630)	0.576	0.720 (0.524-0.857)	0.500 (0.345-0.655)	0.590	0.714 (0.529-0.847)	0.556 (0.396-0.705)	0.625	0.895 (0.686-0.971)	0.472 (0.320-0.630)	0.618
Nodule	0.516 (0.472-0.559)	0.206 (0.185-0.229)	0.295	0.402 (0.373-0.432)	0.328 (0.302-0.354)	0.361	0.373 (0.343-0.404)	0.282 (0.258-0.307)	0.321	0.366 (0.337-0.396)	0.290 (0.266-0.315)	0.324
Pleural thickening	0.667 (0.530-0.780)	0.264 (0.195-0.346)	0.378	0.213 (0.169-0.267)	0.442 (0.359-0.528)	0.288	0.207 (0.159-0.264)	0.364 (0.286-0.450)	0.264	0.233 (0.186-0.288)	0.473 (0.389-0.559)	0.312
Pneumatois	0.667 (0.300-0.903)	0.308 (0.127-0.576)	0.421	0.444 (0.189-0.733)	0.308 (0.127-0.576)	0.364	0.286 (0.082-0.641)	0.154 (0.043-0.422)	0.200	0.333 (0.121-0.646)	0.231 (0.082-0.503)	0.273
Pneumothorax	0.556 (0.373-0.724)	0.484 (0.320-0.652)	0.517	0.600 (0.387-0.781)	0.387 (0.237-0.562)	0.471	0.739 (0.535-0.875)	0.548 (0.378-0.708)	0.630	0.625 (0.453-0.771)	0.645 (0.469-0.789)	0.635
Postoperative Metal	0.840 (0.715-0.917)	0.955 (0.849-0.987)	0.894	0.857 (0.733-0.929)	0.955 (0.849-0.987)	0.903	0.976 (0.877-0.996)	0.932 (0.818-0.977)	0.953	0.913 (0.797-0.966)	0.955 (0.849-0.987)	0.933
Venipuncture	0.786 (0.524-0.924)	0.733 (0.480-0.891)	0.759	0.867 (0.621-0.963)	0.867 (0.621-0.963)	0.867	0.867 (0.621-0.963)	0.867 (0.621-0.963)	0.867	0.857 (0.601-0.960)	0.800 (0.548-0.930)	0.828
Mean	0.681	0.495	0.557	0.618	0.521	0.554	0.620	0.543	0.573	0.627	0.543	0.573

5. There are already large-scale open-source datasets of chest radiograph up to 224316 chest radiographs (MIMIC-CXR) and 174 radiologic findings (PADChest). All 165988 chest radiographs with 14 abnormalities represented by box segmentation constructed in this study should be another good open-source dataset if it is opened to public. Did the investigators develop this dataset for public use?

Response: Thanks for your excellent comments.

MiMIC-CXR, PADChest, ChestXpert and Chest Xray 14 are all valuable public CXR datasets. These datasets have markedly promoted the application of deep learning classification networks to CXRs. However, the above open CXR datasets have essentially no bounding boxes containing localization information of abnormalities and thus cannot meet the localization network requirements. The proposed development concept for AI diagnosis is to train a model that can simultaneously identify and localize multiple abnormalities in CXRs, which is similar to the needs in real clinical settings. Therefore, we spent considerable time and effort in constructing the CXR-AL14 dataset. We are very willing to make the CXR-AL14 dataset available to the public, but the release of the dataset requires approval from hospitals, universities, and higher authorities. We have applied to the relevant departments; however, the approval process is complex due to the impact of COVID-19. We will make this dataset available for public use once approval is available.

6. Development of the X-ray Eye software in the manuscript is not appropriate because it is not appropriately described in purpose, method, and result with appropriate hypothesis, validation, and results for automatic transmission of images from the PACS to the software and priority prompt given for pneumothorax or pneumatosis, which could significantly reduce the time to obtain reported results according to the assertion of the authors.

Response: Highly appreciate your excellent comment.

Just as you said, the detailed results related to the priority prompt given and time reduction were not provided in the original manuscript, and related research has not been performed in the study. Therefore, we deleted the corresponding description, such as the priority prompt given for pneumothorax or pneumatosis, which could significantly reduce the time to obtain reported results.

However, the ultimate goal of this study is to develop a one-stop AI diagnostic tool for CXR interpretation and apply it in routine clinical practice. For medical AI, the visualization of deep learning models is a critical step for clinical application. Therefore, to preliminarily validate the clinical feasibility of the proposed framework, X-ray eye software was developed as a workflow for radiologists in this study. Prospective test also showed the proposed framework owned good performance and clinical application prospects. Therefore, in this paper, X-ray eye software development was described in brief and was just to demonstrate the feasibility and prospects of clinical application of the proposed framework.

(Continued on the next page)

7. In discussion section, It would better describe detailed comparison of performance and what is incremental advance of this study results upon previous studies which has been published already with large-scale datasets.

Response: We appreciate your suggestion.

We reviewed the literature and found relatively few studies that achieved simultaneous localization of multiple abnormalities in CXRs using deep learning localization networks. These localization networks require a CXR dataset with a large number of bounding boxes for multiple abnormalities. Unfortunately, such a dataset is not available for existing public CXR datasets. Although several public CXR datasets exist, nearly all the label information of the dataset is obtained through NPL. Such datasets are typically used with classification networks. However, the localization of abnormalities is more meaningful for interpreting CXRs in clinical practice, and most recent studies have attempted to localize abnormalities using heatmaps generated by classification networks.

The most important result of this study is the creation of the CXR-AL14 dataset, which provides considerable data support for deep learning localization tasks and makes the concept of simultaneous localization of multiple abnormalities in CXRs a reality. Another highlight of our work is that our framework shows good performance and clinical applicability, and the mAP achieved by the YOLOX model reached 0.629 with an IoU-T of 0.5. The performance of the YOLOX model is obviously superior to that of the models trained on a small dataset, such as the VinDr-CXR dataset. We have added the relevant content to the Discussion section of the revised manuscript.

8. Prospective validation of the performance of the YOLOX model should be prospective fashion regarding methodology and should be described in more detail.

Response: Thank you for your suggestion.

After the X-ray eye software was developed, we tested the software for a period of time (January 2022 to May 2022) to determine its ability to visualize the proposed framework. A total of 1022 CXRs were interpreted by the software according to exclusion criteria and collected during this period. Then, these CXRs were annotated by an expert group to achieve GT bounding boxes, and each CXR was manually annotated by two expert radiologists. If two experts could not reach an agreement on the abnormality bounding box, a third expert was invite to discuss to reach a final conclusion. Finally, we compared the bounding boxes generated by the framework with the GT bounding boxes annotated by the experts to obtain the prospective validation results of the YOLOX model. More details on the prospective validation process have been added to the corresponding part in the revised manuscript.

9. In CTR measurement, the maximal distance from the inner margin of the ribs at any level is commonly conducted in clinical practice not at the level of the dome of the right hemidiaphragm as used in this study.

Response: Thank you for this good comment.

To the best of our knowledge, there are currently two methods to measure CTR. In one

method, CTR is the ratio of maximal horizontal cardiac diameter to maximal horizontal thoracic diameter (inner edge of ribs/edge of pleura) (**Figure R4-3A**), and $CTR > 0.50$ is usually considered as cardiomegaly in the method. While, in professional reference book [1], CTR is defined as the ratio of the cardiac diameter (the horizontal distance between the most rightward and most leftward margins of the cardiac shadow) to the thoracic diameter (the distance from the inner margin of the ribs at the level of the dome of the right hemidiaphragm), (**Figure R4-3B**), and $CTR > 0.55$ is usually considered as cardiomegaly. In our studies, the automatic calculation of CTR in the framework was developed by referring to the latter method with the aim to keep consistent with professional reference books. In fact, for the automatic calculation of CTR, the latter method is more difficult to implement than the former. The former method only requires segmentation of both lungs and heart. While for the latter method, the dome point of the right hemidiaphragm must be identified in addition to the segmentation of the double lungs and heart. We added this content to the Discussion section.

Reference

[1] Pouraliakbar, H. Chapter 6 - Chest Radiography in Cardiovascular Disease. in Practical Cardiology (Second Edition) (eds. Maleki, M., Alizadehasl, A. & Haghjoo, M.) 111-129 (Elsevier, 2022).

Figure R4-3 Two methods to calculate the CTR on CXRs.

In method **A**, $CTR = L_1/L$, and a measurement $CTR > 0.50$ is typically identified as abnormal (cardiomegaly).

In method **B**, the CTR is calculated by measuring the thoracic diameter as the distance from the inner margin of the ribs at the level of the dome of the right hemidiaphragm and the cardiac diameter as the horizontal distance between the most rightward and leftward margins of the cardiac shadow. In this case, $CTR = (L_1 + L_2)/L$, and a CTR of less than 0.55 is considered normal in PA CXRs.

(Continued on the next page)

10. Chest radiograph is routinely obtained with posterior anterior projection (PA), but in intensive care unit or for severely ill patients anterior-posterior projection (AP) is usually used to get chest radiograph. CTR cannot be accurately measured in Chest AP.

Response: We highly agree with your comment.

As you said, the CTR cannot be accurately measured in anterior-posterior (AP) CXRs. This issue was considered before we designed the study. The primary reasons for this issue are as follows. First, the heart is far from the flat panel detector (FPD) in the anterior-posterior CXR, which leads to an amplification effect of the heart and affects the measurement of heart transverse diameter. Second, ICU patients typically have bedside CXRs taken in the supine position, and the source-to-film distance is short (100-120 cm). The supine position causes the dome of the diaphragm to rise and the heart to be in a transverse position. For these reasons, anterior-posterior CXRs were excluded from this study. We added corresponding information to the revised manuscript.

REVIEWER COMMENTS

Reviewer #1 (Remarks to the Author):

I thank the authors for the detailed responses. Most of my concerns are properly addressed. Here, I list a few that remain.

1. Will it be better to rephrase "expert" as senior radiologists? otherwise, junior radiologists? please properly define "expert", "radiologist", "senior radiologist", and "junior radiologist" in both the annotation and testing phases. May group them into three classes with years of experience? It will be helpful if a consistent standard is set. Otherwise, it will be less convincing when claiming a superior performance on either side.
2. As shown in Table R1-4, the bounding box inconsistency for the small region and lesions is pretty high. How is it resolved?
3. I will suggest reporting the results of Fast R-CNN or Retina (two-stage ones) in the manuscript as well.

Again, I want to emphasize the contribution of this work mainly lies in the dataset release, which is a sound addon (not unique though, there is another one for the detection task, i.e., VinDr-CXR) to the several existing large-scale chest X-ray public datasets.

Assessment from R#1 on the responses to Reviewer #4:

Comment 2:

Reviewer 4 required a subjective evaluation of the localization accuracy in the external validation process. I do not think the authors address that concern directly. The authors state that they utilized $IoU-T=0.1$ to measure the localization performance quantitatively, and the algorithm performs better than a senior radiologist. I believe it is not proper to have such a conclusion since $IoU-T=0.1$ is quite a low criterion of matching two bounding boxes (roughly 10% of interactions of two boxes will be treated as a correct detection) and favors the algorithm more than common visual inspection.

Comment 5:

Making the dataset publicly accessible is a critical contribution of this work (requiring confirmation) while the authors are working on it.

Comment 6:

I agree with reviewer 4 that the x-ray eye software seems to be an extra part that is not the focus of this scientific paper and is not sufficiently discussed and justified. The authors fail to address this. I will suggest removing the relevant parts of the software.

Comment 7:

The authors did not perform a comparative study of other prior arts on the proposed dataset. Reviewer 4's concern remains.

Comment 8:

The authors added results for the "Prospective validation of the performance", while I believe the "details" required by the reviewer are not only about more results but more description of how the validation is performed, which is still not sufficient.

Reviewer #2 (Remarks to the Author):

The authors have provided reasonable responses to reviewer concerns. I am happy with those. My remaining concerns are as follows:

1. There is no "manufacturer information in Supplementary Table 1". I was expecting to see information about imaging device, version etc. It has been shown that DL algorithms are sensitive to imaging device and acquisition protocols.

2. labeling tool link provided:

407 Both the category and localization of each abnormality in CXRs were annotated by the Labelling
408 tool (v1.8.0, <https://tzutalin.github.io/labellmg/>).

The link on line 408 above is not good.

Also, I do not see information about making the dataset available clearly specified. I do see it in the data statement. I expect that the authors or the journal will make that statement (or some version of it) as part of the publication process.

Reviewer #3 (Remarks to the Author):

I have still have some concerns on the utilization and applicability of this method. At first, there is a lack of novelty issues. In addition, too small CXR size could lead to low F1 score of nodule, which is one of the most important findings in screening setting of CXR.

RESPONSES TO REVIEWER COMMENTS

Please note that the original comments are in a red font, and our responses are in a black font.

Reviewer #1 (Remarks to the Author):

I thank the authors for the detailed responses. Most of my concerns are properly addressed. Here, I list a few that remain.

Response: Thank you very much for your positive comment. Regarding your remaining concerns, we have revised the manuscript in detail. We believe that the manuscript has been further improved.

1. Will it be better to rephrase "expert" as senior radiologists? otherwise, junior radiologists? please properly define "expert", "radiologist", "senior radiologist", and "junior radiologist" in both the annotation and testing phases. May group them into three classes with years of experience? It will be helpful if a consistent standard is set. Otherwise, it will be less convincing when claiming a superior performance on either side.

Response: Thank you for your valuable comments and suggestions.

We are very sorry that we did not clearly define "expert", "senior radiologist", and "junior radiologist". Just as your suggestion, we have grouped them into three classes based on their years of experience in the study. The detailed definitions of radiologists with three levels (expert radiologist, senior radiologist and junior radiologist) and their corresponding tasks involved in this study are listed in **Table R1-1**. We have also clearly expressed the meanings of the three radiologist levels (yellow highlighted font) in the revised manuscript.

Table R1-1 The definitions of radiologists and their tasks involved in this study

Groups of radiologists	Years of experience	The tasks involved in this study
Expert radiologist (Six in total)	more than 20 years	During the construction of the CXR-AL14 dataset, six expert radiologists collectively annotated 8,000 CXRs to train a preliminary model. (P18, line14) During the annotation process of the CXR-AL14 dataset with the help of the human-in-the-loop approach, six expert radiologists finally checked and confirmed the annotations of each CXR. (P18, line21) For all test datasets, three of the six expert radiologists participated in the process of forming the GT annotations without the help of the human-in-the-loop approach. (P21, line24 and P22, line1)
Senior radiologist (Sixteen in total)	more than 10 years and less than 20 years	During the construction of the CXR-AL14 dataset, twelve senior radiologists participated in the first check of the annotations generated by the human-in-the-loop approach. (P18, line19) In the human-machine comparison test, another three senior radiologists were involved in the interpretation of each CXR. (P21, line13) In addition, another senior radiologist participated in the development and validation of the automatic CTR calculation as mentioned in the manuscript. (P22, line17 and P24, line5)
Junior radiologist (Three in total)	6-8 years	In the human-machine comparison test, three junior radiologists were involved in the interpretation of each CXR. (P21, line14) In addition, two of the three junior radiologists participated in the development and validation of the automatic CTR calculation. (P22, line14 and P24, line1)

2. As shown in Table R1-4, the bounding box inconsistency for the small region and lesions is pretty high. How is it resolved?

Response: Thank you for your good question.

We strongly agree with you that it is important to improve the consistency among expert radiologists. When we evaluated the intervariability of six expert radiologists, we also found that the number of inconsistent nodules in “Table R1-4” was slightly higher than that of the other inconsistent abnormalities.

We have analyzed several possible reasons for this finding based on clinical practice. First, the task of identifying and localizing small abnormalities, such as nodule, is difficult for both the YOLOX model and the radiologists. Second, CXRs are overlapping images, so a large degree of overlap between the anatomical structures makes small lesions difficult to observe. For example, it is very easy to produce a missed diagnosis if the nodule is located in the overlapping area of the heart shadow. Third, the vascular cross section looks very much like a nodule in CXRs.

It should be pointed out that some measures have been taken in our research to address this issue to the greatest extent possible. First, expert radiologists need to make full sense of three general rules for the annotation. Second, the expert radiologists must be trained and annotate demo data according to the annotation principles of each abnormality before providing formal annotations. Last but not least, for suspicious abnormalities, especially small abnormalities or abnormalities with blurry boundaries, one expert radiologist should discuss with another one to determine the final annotations.

3. I will suggest reporting the results of Fast R-CNN or Retina (two-stage ones) in the manuscript as well.

Response: Thank you for your suggestion.

In the first revision, we trained the localization model based on Faster R-CNN using the CXR-AL14 dataset and tested it on the held-out test dataset. Furthermore, we further trained the other localization model based on RetinaNet using the CXR-AL14 dataset, and tested it on the held-out test dataset. All of the above results are shown in Table R1-2 and have also added as *supplementary Table 5* in this revised manuscript. It can be easily found that the performance of these two models was not better than that of the YOLOX model, which is shown in Figure R1-1 (this figure was also added as *supplementary Figure 5*) in the revised manuscript.

(Table R1-2 and Figure R1-1 can be found in the next page)

Table R1-2 The AP value of each abnormality achieved by the Faster R-CNN model, the RetinaNet model and the YOLOX model on the held-out test dataset with different IoU-Ts.

	IoU-T=0.5			IoU-T=0.3			IoU-T=0.1		
	Faster R-CNN	RetinaNet	YOLOX	Faster R-CNN	RetinaNet	YOLOX	Faster R-CNN	RetinaNet	YOLOX
Atelectasis	0.275	0.028	0.152	0.318	0.145	0.184	0.318	0.153	0.184
Calcification	0.061	0.637	0.725	0.356	0.792	0.853	0.609	0.859	0.913
Consolidation	0.470	0.455	0.599	0.659	0.615	0.747	0.734	0.683	0.785
Effusion	0.321	0.499	0.627	0.699	0.783	0.839	0.864	0.875	0.922
Emphysema	0.744	0.731	0.681	0.744	0.731	0.681	0.744	0.731	0.681
Fibrosis	0.143	0.431	0.601	0.337	0.567	0.707	0.422	0.605	0.740
Fracture	0.014	0.637	0.819	0.075	0.713	0.869	0.108	0.725	0.875
Mass	0.531	0.546	0.718	0.613	0.592	0.731	0.666	0.623	0.751
Nodule	0.006	0.474	0.348	0.035	0.495	0.614	0.095	0.503	0.695
Pleural thickening	0.178	0.205	0.407	0.334	0.337	0.544	0.387	0.376	0.568
Pneumatoxis	0.152	0.214	0.551	0.379	0.536	0.752	0.518	0.706	0.838
Pneumothorax	0.479	0.579	0.792	0.595	0.745	0.903	0.693	0.812	0.939
Postoperative Metal	0.418	0.875	0.867	0.739	0.943	0.896	0.850	0.958	0.902
Venipuncture	0.008	0.777	0.916	0.037	0.856	0.955	0.108	0.887	0.970
mAP	0.271	0.506	0.629	0.423	0.632	0.734	0.508	0.678	0.769

Figure R1-1 Performance comparison between the Faster R-CNN model, the RetinaNet model and the YOLOX model on the held-out test dataset with different IoU-Ts.

Again, I want to emphasize the contribution of this work mainly lies in the dataset release, which is a sound add-on (not unique though, there is another one for the detection task, i.e., VinDr-CXR) to the several existing large-scale chest X-ray public datasets.

Response: Thank you for your positive comments.

After our efforts, we made the CXR-AL14 dataset available for public use with the

approval of our institution and uploaded it to the website (cxr-al14.top). It should be noted that the CXR-AL14 dataset will only be available for academic research (ie, as a reference for model parameters and study designs), and not for other purposes (ie, commercial use). Interested researchers need to register their personal and institutional information on this website (cxr-al14.top) and send data access requests to the web administrator. The web administrator will review the requests for consideration. Once approved, the dataset can be downloaded. Note that interested researchers who have utilized the CXR-AL14 dataset for research must cite this article.

Assessment from R#1 on the responses to Reviewer #4:

Comment 2:

Reviewer 4 required a subjective evaluation of the localization accuracy in the external validation process. I do not think the authors address that concern directly. The authors state that they utilized IoU-T=0.1 to measure the localization performance quantitatively, and the algorithm performs better than a senior radiologist. I believe it is not proper to have such a conclusion since IoU-T=0.1 is quite a low criterion of matching two bounding boxes (roughly 10% of interactions of two boxes will be treated as a correct detection) and favors the algorithm more than common visual inspection.

“The original Comment 2 from Reviewer 4: IOU-threshold is a usual method to evaluate the accuracy of YOLOX model. However, qualitative evaluation of lesion detection on chest radiograph by expert radiologists might be more accurate method to evaluate the performance of the model. It could be done on external validation process.”

Response: Thank you for your meaningful comments.

We greatly agree with two of the reviewers. Based on the original comment of Reviewer #4, we performed a qualitative evaluation of the YOLOX model on the external test dataset which was constructed by the four external hospitals (a total of 10,945 CXRs). It should be noted that the annotations (including the category and localization of the bounding box for each abnormality) of the external test dataset were formulated by the expert radiologists. When an IoU-threshold was not considered (as the comments of Reviewer #4), the qualitative evaluation of lesion detection on CXRs by expert radiologists is essentially a simple classification problem for the YOLOX model, which can be evaluated by ROC curves. The results obtained by the YOLOX model in the qualitative evaluation conducted on the external test dataset are given in **Figure R1-2**, which includes the ROC curve and AUC value for each abnormality and no findings, where each ROC curve represents the ability of the model to identify the corresponding abnormality on a CXR. Note that simply considering classification is much simpler than considering both classification and localization. As can be seen, the YOLOX model achieved excellent performance in terms of identifying most abnormalities, but its performance was relatively low only for nodule identification, which was consistent with the evaluation results obtained with IoU-T.

Note that the aim of this study was to simultaneously identify and localize multiple abnormalities on CXRs. This qualitative evaluation approach is not very suitable for this study, so we did not add it to the revised manuscript.

(**Figure R1-2** can be found in the next page)

Figure R1-2 The ROC curves of each abnormality for conducting a qualitative evaluation of the YOLOX model on the external test dataset.

In this study, we used a professional evaluation system from the field of object detection to evaluate the performance of the YOLOX model, considering both the accuracy of localization (IoU-T) and the accuracy of classification (PR curves). To address the comment from Reviewer #1 on the responses to Reviewer #4, we explore this issue from a different perspective for evaluating the localization accuracy. As we know, the IoU value must be used if we want to accurately evaluate the accuracy of localization. Therefore, under the premise of correct category predictions, the IoU was used as a final evaluation indicator to reflect the identification and localization performance of the YOLOX model, as this metric can reflect a subjective evaluation to some extent. On the premise of correctly identifying categories, we calculated the average IoU value for each abnormality (in comparison with the GT bounding boxes annotated by expert radiologists) in order to demonstrate the accuracy of the YOLOX model for localizing abnormalities, which can be seen in **Table R1-3**. It is easy to see from **Figure R1-2** and **Table R1-3** that the YOLOX model achieved excellent identification and

localization performance for multiple abnormalities. We reported this result in the revised manuscript (Table R1-3 was added as *supplementary Table 6*).

Table R1-3 Average IoU value of each abnormality achieved by the YOLOX model on the external test dataset (a total of 10,945 CXRs from four multicentres).

Abnormality (category)	Average IoU (mean±SD)
Atelectasis	0.590±0.253
Calcification	0.762±0.175
Consolidation	0.714±0.195
Effusion	0.550±0.146
Emphysema	0.917±0.037
Fibrosis	0.686±0.196
Fracture	0.769±0.146
Mass	0.792±0.166
Nodule	0.658±0.212
Pleural thickening	0.661±0.174
Pneumatoxis	0.458±0.271
Pneumothorax	0.734±0.175
Postoperative metal	0.841±0.154
Venipuncture	0.800±0.111

Comment 5:

Making the dataset publicly accessible is a critical contribution of this work (requiring confirmation) while the authors are working on it.

Response: Thank you for your comment.

After our efforts, we made the CXR-AL14 dataset available for public use with the approval of our institution and uploaded it to the website (cxr-al14.top). It should be noted that the CXR-AL14 dataset will only be available for academic research (ie, as a reference for model parameters and study designs), and not for other purposes (ie, commercial use). Interested researchers need to register their personal and institutional information on this website (cxr-al14.top) and send data access requests to the web administrator. The web administrator will review the requests for consideration. Once approved, the dataset can be downloaded. Note that interested researchers who have utilized the CXR-AL14 dataset for research must cite this article.

Comment 6:

I agree with reviewer 4 that the x-ray eye software seems to be an extra part that is not the focus of this scientific paper and is not sufficiently discussed and justified. The authors fail to address this. I will suggest removing the relevant parts of the software.

Response: Thank you for your suggestion. We have removed the relevant parts of the

software (font with track changes) in the revised manuscript.

Comment 7:

The authors did not perform a comparative study of other prior arts on the proposed dataset. Reviewer 4's concern remains.

Response: Thank you for your comment. We further used other prior localization networks, including Faster R-CNN and RetinaNet, to train the deep learning model, and found that the performance of the YOLOX model was the best one. More details can be found in **Figure R1-1**. We have added this result into the revised manuscript (*supplementary Figure 5*).

Figure R1-1 Performance comparison between the Faster R-CNN model, the RetinaNet model and the YOLOX model on the held-out test dataset with different IoU-Ts.

Comment 8:

The authors added results for the "Prospective validation of the performance", while I believe the "details" required by the reviewer are not only about more results but more description of how the validation is performed, which is still not sufficient.

Response: Thank you for your suggestions.

We added more description of how the prospective validation is performed as follows:

A prospective validation was conducted on the 1022 CXRs to test the clinical applicability of the YOLOX model. After the YOLOX model was developed, we used this model to interpret the newly generated CXRs in clinical practice. Then, two expert radiologists annotated the GT bounding boxes for these CXRs without knowing the results achieved by the YOLOX model. Finally, the prospective validation results could be obtained for the YOLOX model by comparing the predicted bounding boxes with the GT bounding boxes.

These contents were added in **Method section 4.6** (yellow highlighted font) in the revised manuscript.

Reviewer #2 (Remarks to the Author):

The authors have provided reasonable responses to reviewer concerns. I am happy with those. My remaining concerns are as follows:

Response: Thank you very much for your positive comment. Regarding your remaining concerns, we have revised the manuscript in detail. We believe that the manuscript has been further improved.

1. There is no "manufacturer information in Supplementary Table 1". I was expecting to see information about imaging device, version etc. It has been shown that DL algorithms are sensitive to imaging device and acquisition protocols.

Response: Thank you for your valuable comments and suggestions. We have added the device information of all the CXRs involved in this study, and updated *Supplementary Table 1* in the supplementary materials.

Supplementary Table 1 Manufacturer and device information concerning the CXRs in each dataset.

Manufacturer information	Device information	CXr-AL14 dataset		Held-out test dataset	External dataset A	External dataset B	External dataset C	External dataset D	Prospective dataset
		Training dataset	Tuning dataset						
Shimadzu	RADspeed Pro 50	0	0	0	9	2628	2	0	0
Carestream Health	DRX Evolution	8685	947	415	9	0	718	0	262
KODAK	DirectView DR7500	12286	1423	544	0	0	0	0	413
	DirectView DR3500	0	0	0	2371	0	0	0	0
	DirectView DR3000	0	0	0	0	0	1929	0	0
SIEMENS	AXIOM Aristos VX Plus	128454	14193	5041	0	0	0	0	347
	Multix Fusion Max	0	0	0	589	0	0	0	0
DDIT	WV3000T	0	0	0	0	5	0	0	0
Mindray	DigiEye 680	0	0	0	0	0	2	0	0
General Medical Merate S.p.A.	CALPYSO	0	0	0	0	0	0	2683	0
Total		149425	16563	6000	2978	2633	2651	2683	1022

2. labeling tool link provided:

407 Both the category and localization of each abnormality in CXRs were annotated by the Labellmg

408 tool (v1.8.0, <https://tzutalin.github.io/labellmg/>).

The link on line 408 above is not good. (Response can be found in the next page)

Response: Thank you very much for your comment. We apologize that the download address is unavailable in the original manuscript. We have corrected the download address in the revised manuscript: <https://pypi.org/project/labelImg/1.8.0/>.

Also, I do not see information about making the dataset available clearly specified. I do see it in the data statement. I expect that the authors or the journal will make that statement (or some version of it) as part of the publication process.

Response: Thank you for your suggestions. **Data availability** is mentioned in our cover letter and reporting summary for the manuscript. We will update **data availability** in subsequent publication process. Thank you again for your reminding.

Reviewer #3 (Remarks to the Author):

I have still have some concerns on the utilization and applicability of this method. At first, there is a lack of novelty issues. In addition, too small CXR size could lead to low F1 score of nodule, which is one of the most important findings in screening setting of CXR.

Response: Thank you for your valuable comments.

The utilized method (YOLOX algorithm) is a classic deep learning-based localization network. No optimization or improvement was performed on the YOLOX algorithm in our study. However, the purpose of this work was to simultaneously identify and localize multiple abnormalities on CXRs which is urgently required radiologists in real clinic work. Our research made the concept of simultaneously identifying and localizing multiple abnormalities on CXRs a reality. Additionally, during the process of the study, the CXR-AL14 dataset, which is the largest CXR dataset with multi-abnormality GT bounding boxes in the world, was also created. We believe that the CXR-AL14 dataset can provide great data support for novel explorations of deep learning-based algorithms, which is also our future research work.

In addition, in terms of calculating the cardiothoracic ratio, based on the precise segmentation of both lungs and the heart, we proposed corresponding algorithm to calculate the thoracic diameter and the cardiac diameter, thereby accurately calculating the cardiothoracic ratio according to professional reference book.

We strongly agree with you that too small CXR size could lead to a low F1-score of nodule. A small CXR size leads to an image resolution reduction, and the bounding box sizes of nodules correspondingly decreases. Identifying and localizing small abnormalities, such as nodules, is a difficult task for both deep model and radiologists. Thus, too small CXR size could lead to a low F1-score of nodule.

REVIEWERS' COMMENTS

Reviewer #1 (Remarks to the Author):

All my previous concerns have been adequately addressed in the revision. Therefore, I do not have any further comments. The authors also provided the link for downloading the large-scale dataset, which I believe would significantly contribute to the community.

Reviewer #2 (Remarks to the Author):

The authors have diligently and comprehensively responded to comments from previous reviews. I have no further concerns.

Reviewer #3 (Remarks to the Author):

Mainly, CXR is used for nodule detection in actual clinical setting. However this study don't show any meaningful progress in this issue. Therefore, due to lacks of technical novelty and clinical usefulness from small CXR size, I don't think that this study is recommended for publication in this journal.

RESPONSES TO REVIEWER COMMENTS

Please note that the original comments are in a red font, and our responses are in a black font.

Reviewer #1 (Remarks to the Author):

All my previous concerns have been adequately addressed in the revision. Therefore, I do not have any further comments. The authors also provided the link for downloading the large-scale dataset, which I believe would significantly contribute to the community.

Response: We gratefully thank you for your constructive remarks and useful comments, which has significantly improved the quality of the manuscript.

Reviewer #2 (Remarks to the Author):

The authors have diligently and comprehensively responded to comments from previous reviews. I have no further concerns.

Response: Thank you very much for your approval of our manuscript. We would also thank you for your valuable and insightful comments, which are very helpful to the improvement of our manuscript.

Reviewer #3 (Remarks to the Author):

Mainly, CXR is used for nodule detection in actual clinical setting. However this study don't show any meaningful progress in this issue. Therefore, due to lacks of technical novelty and clinical usefulness from small CXR size, I don't think that this study is recommended for publication in this journal.

Response: Thank you very much for your comment.

In general, CXRs are used to diagnose multiple abnormalities in clinical practice as the types listed in this manuscript, not just nodules. The purpose of the study is to accurately identify and localize multiple abnormalities and calculate CTR, not to only detect the nodule.

In fact, the value of CXR to detect nodule (especially small size) is limited in actual clinical practice due to the reasons given as following: CXRs are overlapping images, vascular sections are often misdiagnosed as nodules. Compared with CXR, CT images have great advantage in the detection of

pulmonary nodules. As we know, CT images can detect the small nodule with 2-3mm in diameter which is not possible to be detected in CXRs. Therefore, CT is the preferred method for detecting nodules.

In clinical works, small-size CXR not only affects the detection of pulmonary nodules, but also has a great impact on that of other abnormalities. Therefore, small size CXR is not commonly used to detect abnormality in order to make sure accurate detection in actual clinical works. Comprehensive consideration of factors in computing resources and computing speed, the size of input CXR is set to 1280×1280 in our study. Note that this size is not small for medical images processing [1,2,3].

The biggest highlight of our work is the creation of a large CXR dataset with huge number of bounding boxes for multiple abnormalities, named CXR-AL14. Based on the above dataset, the YOLOX algorithm was used to develop a deep model that can simultaneously identify and locate 14 abnormalities in CXRs. As we know, the CXR-AL14 dataset is the largest CXR dataset with GT bounding boxes in the world, which would significantly promote the research of the localization task in CXRs. The framework in the study is not only more suitable for routine clinical work than AI model reported previously, but also provides good performance, generalization ability and strong clinical application prospect. In addition, by using deep learning technology, we proposed an automatic calculation method to calculate the CTR based on deep learning, which achieves accurate measurement.

Finally, we think that our manuscript is particularly suitable for *Nature Communications*. First, this study involved interdisciplinary research. Accurate identification and localization of multiple abnormalities and the calculation of the CTR with one framework can truly meet the daily clinical needs of radiologists. Second, we constructed a large dataset of CXRs with bounding boxes using the human-in-the-loop method. We believe that the creation of this dataset may promote deep learning-based research on the localization of multiple abnormalities. Thus, this study is of interest to deep learning researchers, clinical radiologists, chest-related clinicians, and emergency department doctors. *Nature Communications* is a highly influential and comprehensive multidisciplinary journal, and articles in this journal are open access and can be downloaded for free. Therefore, we think that our manuscript is very suitable for *Nature Communications*, an international journal, and will be of great interest to its readers.

References

[1] Kim Y G , Lee S M , Lee K H ,et al.Optimal matrix size of chest radiographs for computer-aided detection on lung nodule or mass with deep learning[J].European Radiology, 2020.DOI: 10.1007/s00330-020-06892-9.

[2] Wang G , Liu X , Shen J ,et al.A deep-learning pipeline for the diagnosis and discrimination of viral, non-viral and COVID-19 pneumonia from chest X-ray images[J].Nature Biomedical Engineering[2023-07-03].DOI: 10.1038/s41551-021-00704-1.

[3] Li X , Shen L , Xie X ,et al.Multi-resolution Convolutional Networks for Chest X-Ray Radiograph Based Lung Nodule Detection[J].Artificial Intelligence in Medicine, 2019, 103(Apr): 101744.DOI: 10.1016/j.artmed.2019.101744.